# A general one-step protocol to generate impermeable fluorescent HaloTag substrates for in situ live cell application and super-resolution imaging

Kilian Roßmann [1], Ulrich Pabst [1], Bianca C. Baciu [1], Siqi Sun [1], Christiane Huhn [1], Christina Holmboe Olesen [1], Maria Kowald [1], Eleni Tapp [1], Marie Bieck[1], Ramona Birke[1], Brenda C. Shields [2], Pyeonghwa Jeong [3], Jiyong Hong [3], Michael R. Tadross [2], Joshua Levitz [4], Martin Lehmann [1], Noa Lipstein [1] & Johannes Broichhagen [1] ✉

Visualization of proteins can be achieved by genetically grafting HaloTag Protein (HTP) into the protein of interest followed by incubation with a dye-linked HaloTag Ligand (HTL). This approach allows for use of fluorophores optimized for specific optical techniques or of cell-impermeable dyes to selectively label cell surface proteins. However, these two goals often conflict, as many high-performing dyes exhibit membrane permeability. Here we show that several dye-HTL reagents can be made cell-impermeable by inserting a charged sulfonate directly into the HTL, leaving the dye moiety unperturbed, using a one-step protocol. We validate such compounds, termed dye-SHTL (dye shuttle), in living cells, and demonstrate exclusive membrane staining. In transduced primary hippocampal neurons, we label a neuromodulatory receptor with dyes optimized for stimulated emission by depletion super-resolution microscopy, allowing accuracy in distinguishing surface versus internal receptors of the presynaptic terminal. This approach offers broad utility for surface-specific protein labelling.

Self-labelling protein (SLP) tags, e.g., SNAP and HTP, are at the forefront of approaches to label proteins of interest bio-orthogonally with chemical moieties in living cells and tissue[1–3]. Their main application is for fluorescence microscopy, where dyes are covalently attached to SLP fusion proteins to investigate protein localization and dynamics[4,5]. A particularly important super-family of signalling proteins are cell surface receptors, which mediate information transmission between cells. However, standard protocols label both the extracellular and intracellular populations, including proteins in endo/lysosomal compartments

and translated proteins that remain in the endoplasmic reticulum (Fig. 1a, left). Selective labelling of the cell-surface subset can be achieved with cell-impermeable dyes. Typically, this requires that the dye itself bears anionic residues, e.g., carboxylates and sulfonates (Fig. 1a, middle)[6,7]. However, many of the best dyes for single molecule and super-resolution microscopy[8–11] are cell permeable, limiting their utility for selective surface labelling.

Traditionally, dyes are made less cell permeable by custom modification to the dye itself. For instance, Jonker et al., added carboxylates on the 3-position of the azetidine of JaneliaFluor635 (JF$_{635}$),

[1]Leibniz-Forschungsinstitut für Molekulare Pharmakologie (FMP), Berlin, Germany. [2]Duke University, Departments of Neurosurgery, Neurobiology and Biomedical Engineering, Durham, NC, USA. [3]Duke University, Department of Chemistry, Durham, NC, USA. [4]Weill Cornell Medicine, Department of Biochemistry and Biophysics, New York, NY, USA. ✉e-mail: broichhagen@fmp-berlin.de

i. HTP-TM-SNAP

i.e. SNAP-TM-HTP

**b** Poc *et al.*, *Chem. Sci.* 2020
X = O: **SBG-TMR**
X = SiMe₂: **SBG-SiR**

Jonker *et al.*, *J. Cell Sci.* 2020
**JF₆₃₅i-HTL**

Birke *et al.*, *Org. Biomol. Chem.* 2022
X = O: **BG- and CA-Sulfo549**
X = SiMe₂: **BG- and CA-Sulfo646**

**d** this work:
X = O: **TMR-d12**
X = SiMe₂: **SiR-d12**

**SBG**: impermeable SNAP substrate

**HTL**: permeable Halo substrate
(former naming: CA = chloroalkane)

**BG**: permeable SNAP substrate

**SHTL**: impermeable Halo substrate

X = O: **TMR-d12-HTL**
X = SiMe₂: **SiR-d12-HTL**

X = O: **TMR-d12-SHTL**
X = SiMe₂: **SiR-d12-SHTL**

| | $\lambda_{exc/em}$ (nm) | $\Phi$ | $\tau$ (ns) |
|---|---|---|---|
| **TMR-d12-HTL** | 553 / 577 | 0.46 | 2.55 |
| **TMR-d12-SHTL** | 554 / 579 | 0.47 | 2.56 |
| **HTP:TMR-d12** | 554 / 576 | 0.53 | 3.04 |
| **HTP:S-TMR-d12** | 561 / 588 | 0.54 | 3.39 |
| **SiR-d12-HTL** | 654 / 669 | 0.48 | 3.09 |
| **SiR-d12-SHTL** | 650 / 669 | 0.49 | 3.42 |
| **HTP:SiR-d12** | 652 / 668 | 0.54 | 3.62 |
| **HTP:S-SiR-d12** | 655 / 672 | 0.54 | 3.76 |

yielding impermeable and fluorogenic JF₆₃₅i for investigating endocytotic turnover[6] (Fig. 1a, b), and Eiring et al. reported benefits for single-molecule localization microscopy with a highly charged Cy5b[10]. Our own laboratories have addressed this in a similar vein, by fusing a sulfonate on the same azetidine position via amide bond coupling using taurine, effectively converting JF₅₄₉ and JF₆₄₆ to Sulfo549 and Sulfo646 (ref. 7), respectively (Fig. 1a, right and Fig. 1b), which have

been used to study kainate receptor stoichiometry[12] and intracellular trafficking properties of GPCR subtypes[13].

A more general and effective strategy would offer a universal solution for all dyes. For instance, we recently described a modified version of the SNAP ligand in which the benzylguanine (BG) leaving group carries a negative charge due to incorporation of a sulfonate on the 8-position of guanine, resulting in 'SBG substrates' that are

**Fig. 1 | Logic for impermeable dyes to label cell surface proteins. a** Known strategies to address extracellularly exposed self-labelling tags. **b** Chemical structures showing previously introduced anionic charges. **c, d** Our strategy to install a sulfonate charge on the HaloTag Ligand (HTL), as per (**a, b**). **e** Modelling of the HaloTag Protein (HTP) bound to TMR-SHTL. **f** Synthesis of TMR-d12-SHTL and SiR-d12-SHTL. **g** Excitation and emission profiles of TMR-d12 comparing HTL to SHTL conjugates. $n = 3$. **h** In vitro protein labelling of apo-HTP confirms binding by full protein mass spectrometry. $n = 1$ **i** Labelling kinetics of apo-HTP with TMR-d12-HTL and TMR-d12-SHTL. Min-to-max box and whisker with median, for all $n = 9$. TMR-d12-HTL: mean = 49.9 s, min-max = 34.8-67.0 s; TMR-d12-HTL: mean = 55.6 s, min-max = 41.8-76.6 s. 25–75% percentile for both. Unpaired Student's t-test, two-tailed, normal distribution assumption, $p = 0.3442$. **j** Fluorescence quantum yields of TMR/SiR-d12 conjugated to HTL/SHTL with or without HTP-bound. Min-to-max box and whisker with median. TMR-d12-HTL: $n = 10$;

mean = 0.464; min-max = 0.455-0.477, HTP:TMR-d12: $n = 12$; mean = 0.530; min-max = 0.521-0.540, TMR-d12-SHTL: $n = 4$; mean = 0.467; min-max = 0.462-0.472, HTP:S-TMR-d12: $n = 7$; mean = 0.536; min-max = 0.513-0.55, SiR-d12-HTL: $n = 5$; mean = 0.479; min-max = 0.475-0.482, HTP:SiR-d12: $n = 8$; mean = 0.536; min-max = 0.532-0.542, SiR-d12-SHTL: $n = 5$; mean = 0.487; min-max = 0.484-0.491, HTP:S-SiR-d12: $n = 6$; mean = 0.535; min-max = 0.533-0.540. 25-75% percentile for all. **k** Fluorescence lifetimes of the same reagents. Min-to-max box and whisker with median, N = 3 sample preparations with $n = 3$ ROIs each. TMR-d12-HTL: mean = 2.55 ns; min-max = 2.39-2.82 ns, HTP:TMR-d12: mean = 3.04 ns; min-max = 2.72-3.60 ns, TMR-d12-SHTL: mean = 2.56 ns; min-max = 2.35-2.73 ns, HTP:S-TMR-d12: mean = 3.39 ns; min-max = 2.98-3.73 ns; SiR-d12-HTL: mean = 3.09 ns; min-max = 2.88-3.26 ns, HTP:SiR-d12: mean = 3.62 ns; min-max = 3.08-3.96 ns, SiR-d12-SHTL: mean = 3.42 ns; min-max = 3.39-3.46 ns, HTP:S-SiR-d12: mean = 3.76 ns; min-max = 3.51-4.03 ns. 25-75% percentile for all. **l** Table summarizing values from (**g, j, k**).

released upon covalent reaction (Fig. 1a, right and Fig. 1b)[14]. Unfortunately, this approach is not possible for the HaloTag system, since the HTL leaving group is a chloride anion, and chemical oxidation would result in hypochlorite ($RClO$), chlorite ($RClO_2$), chlorate ($RClO_3$) or perchlorate ($RClO_4$), which all would maintain an overall neutral charge. Nevertheless, we pursued this approach and synthesized negatively charged TMR-HTL-$OSO_3^-$ and TMR-HTL-$OPO_3^{2-}$ which bear an alkyl sulfonate and phosphate, respectively, instead of the alkyl chloride as a leaving group (Supplementary Fig. 1a). While the phosphate is unstable over minutes in PBS as assessed by LCMS, the sulfonate did not show decomposition overnight (Supplementary Fig. 1b). Still, and as expected, TMR-HTL-sulfonate did not covalently label recombinant HTP as assessed by full protein mass spectrometry (Supplementary Fig. 1c).

Having explored and ruled out the modified chloride leaving group approach, in this study we develop a simple approach for late-stage introduction of a sulfonate on the amide bond that links HTL to the dye (Fig. 1c). We determine the photophysical properties of the subsequent modified dyes, validate their lack of membrane permeability in cell lines and primary neurons, employ them for (dual color) super-resolution analysis of GPCR nanolocalization in neurons, adapt them for use with a next-generation HTL, cover three different dye classes and devise a protocol to quantitatively convert commercially available JaneliaFluor-HTL reagents—and potentially other dyes—to enable straightforward, widespread application.

## Results
### Launching SHTL: Sulfonation of fluorescent HaloTag Ligands
To produce membrane-impermeable HTL-targeting probes, we aimed to introduce a sulfonate on the amide bond of available dye-HTL conjugates (Fig. 1c, d), giving access to dye-SHTL substrates. We first turned to molecular modelling to assess if the modified linker would be tolerated by the HTP, docking the density functional theory (DFT)-optimized ligands covalently to the apo HTP structure (PDB-5UY1)[15–17]. We found that the added C3-linker bearing the sulfonate does not sterically hamper the exposed dye on the protein surface (Fig. 1e) and does not exhibit biologically significant differences in estimated binding thermodynamics (see SI and Supplementary Fig. 2–6). Next, we synthesized two dye-SHTL substrates starting with our recently reported TMR-d12 and SiR-d12 dyes, the latter has been shown to be an outstanding candidate for STED super-resolution microscopy[18]. Dissolving these dye-HTL substrates in DMF and subsequently adding sodium hydride (NaH 60% in mineral oil) before 1,3 propane sultone, led to clean conversion, yielding TMR-d12-SHTL and SiR-d12-SHTL within an hour in 94% and 91% yield, respectively, after HPLC purification (Fig. 1f) (see SI). Introduction of the charged sulfonate only slightly changed the excitation and emission profiles, as illustrated for TMR-d12 (HTL: $\lambda_{Ex/Em} = 553 / 577$ nm; SHTL: $\lambda_{Ex/Em} = 554 / 579$ nm) (Fig. 1g). Full protein mass spectrometry on recombinant HTP (ref. 7) confirmed that TMR-d12-SHTL and SiR-d12-SHTL bind to HTP with

quantitative stoichiometry (Fig. 1h), with no significant difference in labelling kinetics as determined by fluorescence polarization ($t_{1/2} \sim 50$ sec) (Fig. 1i). We next measured quantum yields (Fig. 1j) and fluorescence lifetimes (Fig. 1k) to test if sulfonation impairs dye photophysics. The same trend in increasing quantum yield upon binding to HTP was observed for all dyes ($\Phi_{HTP\text{-}free} = 47$–49%; $\Phi_{HTP\text{-}bound} \sim 53\%$), while the sulfonated versions displayed up to 11% longer fluorescence lifetimes over non-sulfonated precursors (Fig. 1l).

### Cell imaging validates membrane impermeability of dye-SHTLs
We next tested both dyes in HEK293T cells transiently transfected with our previously reported SNAP-TM-HTP and HTP-TM-SNAP constructs[7]. Having each tag residing on the opposite side of the cellular membrane separated by a single transmembrane (TM) domain enabled us to simultaneously assess membrane permeability and labelling with built-in controls for expression and cell health. We first used HTP-TM-SNAP cells, using the cell-permeable BG-JF$_{646}$ to label all intracellular proteins, co-applied for 30 minutes with either TMR-d12-HTL (Fig. 2a, upper row) or TMR-d12-SHTL (Fig. 2a, lower row). As expected, TMR-d12-HTL showed considerable intracellular signals (Fig. 2b) confirmed by the similarity of line-scan profiles to the BG-JF$_{646}$ intracellular reference (Fig. 2c, raw fluorescence left and normalized right). By contrast, TMR-d12-SHTL (Fig. 2d) exhibited selective surface labelling, with line scans differing considerably from the intracellular reference (Fig. 2e). We then swapped the self-labelling tags with the SNAP-TM-HTP construct, switching to a cell-impermeable BG-SulfoJF$_{646}$ so that the SNAP label serves as a surface-exclusive reference. Consistent with our prior findings, TMR-d12-HTL (Fig. 2f, upper row) exhibited intracellular labelling (Fig. 2g, h), while TMR-d12-SHTL (Fig. 2f, lower row) showed no detectable labelling (Fig. 2i, j). The same experiment was conducted with SiR-d12 fused to HTL (Fig. 2k–o) vs. SHTL (Fig. 2p–t) and respective red fluorophores (BG-JF$_{549}$ and BG-Sulfo549), confirming that SHTL shows no membrane permeability regardless of the fluorophore. We further titrated SiR-d12 HTL and SHTL conjugates up to 5000 nM on HTP-TM-SNAP expressing HEK293T cells, and again observed clear membrane staining for sulfonated versions (Supplementary Fig. 7).

### Expanding the dye-SHTL scaffold and color palette
Rhodamines linked by peptide bonds to SLP ligands are often the dye and connection type of choice for HTP labelling and cellular imaging[3]. However, we wondered if our protocol is more general and amenable to completely distinct scaffolds as a plethora of fluorophores exist with new variants emerging rapidly. Nitrobenzodiazaisooxazole (NBD) are classical fluorophores in the green spectrum with a vastly different molecular structure than rhodamines[19]. They have recently been upgraded to span a larger color spectrum and become known as the SCOTfluors (ref. 20), and furthermore have been used on solid phase support for peptide synthesis[21,22]. Motivated by these properties, we picked this dye class for testing our strategy using amine derivatization

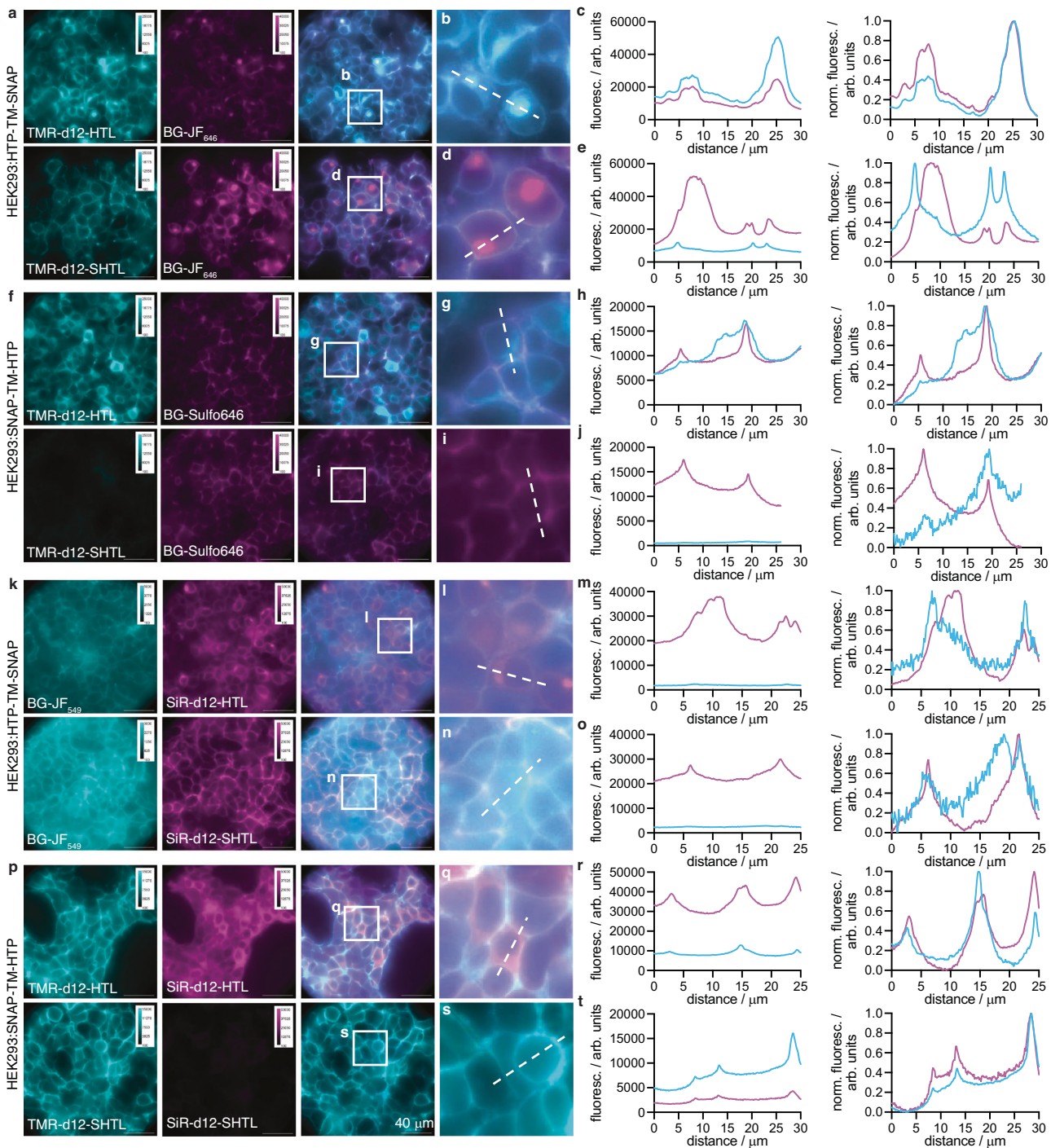

**Fig. 2 | Live cell imaging of dye-SHTL in transfected HE293T cells. a–e** HEK293T expressing HTP-TM-SNAP. Intracellular SNAP labelled with BG-JF$_{646}$; extracellular HTP labelled with TMR-d12-HTL or TMR-d12-SHTL. Widefield imaging (**a**), zoom-ins (**b, d**) and line scans (**c, e**). **f–j** HEK293T cells expressing SNAP-TM-HTP. Extracellular SNAP labelled with BG-SulfoJF$_{646}$; intracellular HTP labelled with TMR-d12-HTL or TMR-d12-SHTL. Widefield (**f**), zoom-in (**g, i**) and line scans (**h, j**). **k–o** As for **a–e** but staining with BG-JF$_{549}$ and SiR-d12-(S)HTL. **p–t** As for **f–j** but staining with BG-SulfoJF$_{549}$ and SiR-d12-(S)HTL. Imaging was performed for all conditions in N = 3 preparations with *n* = 3 images per condition. For all images, scale bar = 40 μm.

(Fig. 3a). Before commencing with the synthesis, we docked NBD-HTL (Fig. 3b) and NBD-SHTL (Fig. 3c) covalently to the HaloTag in silico and observed that both structures are sterically tolerated. NBD-HTL was then straightforwardly synthesized by using NBD-Cl and HTL-NH$_2$ in EtOH to precipitate analytically clean permeable ligand, which was subjected to treatment with NaH in DMF, and sulfonation by propylene sultone to yield NBD-SHTL (Fig. 3d). Since the amine defines the photophysical properties of NBD, we recorded an expected red-shift in

its excitation when bis-alkylated, however, emission remains comparable ($\lambda_{Exc/Em}$ (NBD-HTL) = 479 / 552 nm; $\lambda_{Exc/Em}$ (NBD-SHTL) = 498 / 546 nm) (Fig. 3e). We confirmed that NBD-(S)HTL does label recombinant HTP (Fig. 3f) by full protein mass spectrometry, and then incubated both ligands with HEK293T cells for confocal imaging. While cells uptake NBD-HTL non-specifically, we did not observe any fluorescence signal when using NBD-SHTL (Supplementary Fig. 8). Next, we transfected HEK293T cells with the aforementioned SNAP-TM-HTP

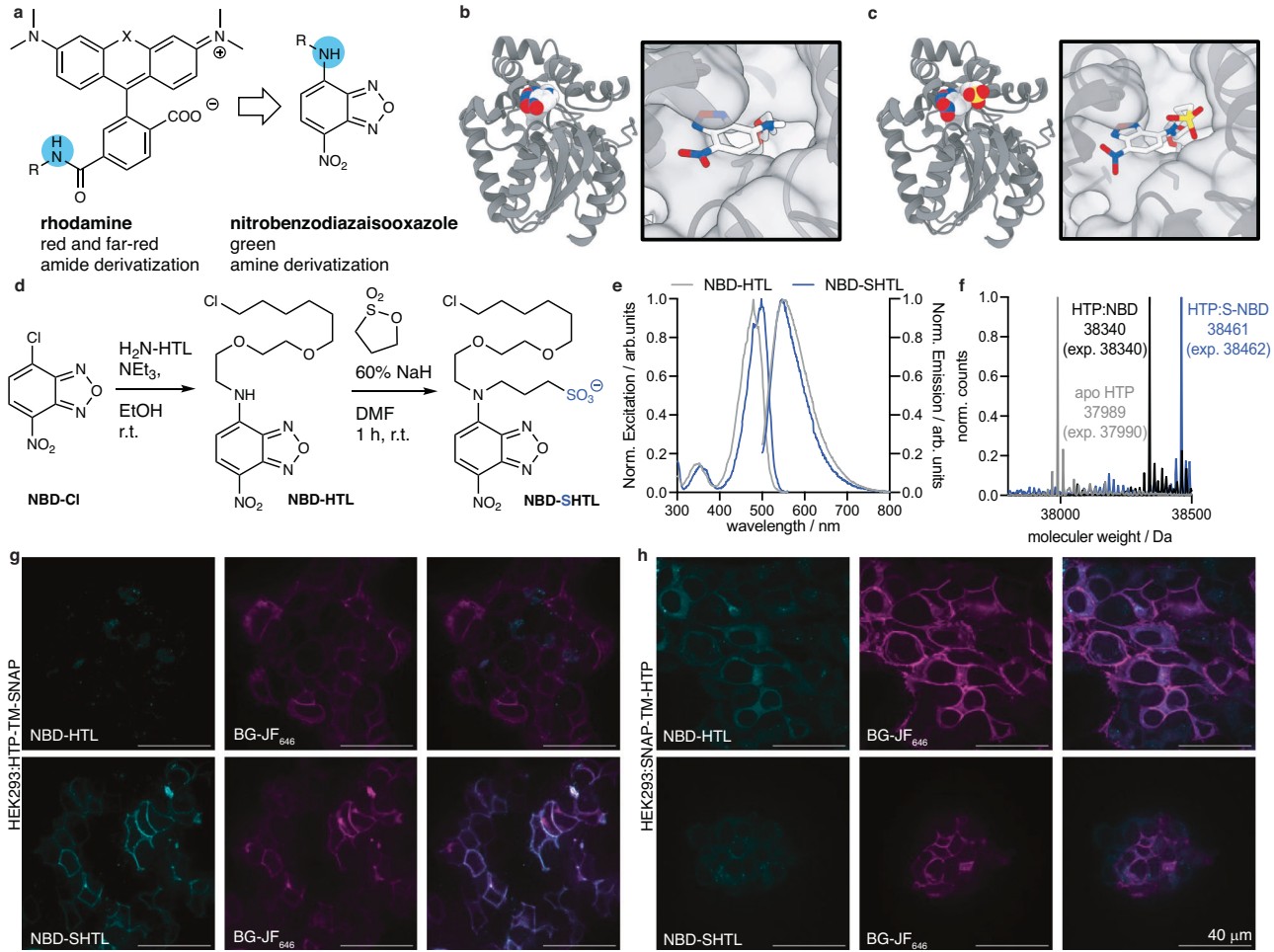

**Fig. 3 | Sulfonation on an amine linked NBD dye. a** Expansion of the approach from red/far-red amide conjugated HTL dyes to green and amine linked NBD. **b**, **c** Modelling of the HaloTag Protein (HTP) bound to NBD-(S)HTL. **d** Synthesis of NBD-HTL and NBD-SHTL. **e** Excitation and emission profiles of NBD-HTL and NBD-SHTL. $n = 3$. **f** In vitro protein labelling of apo-HTP confirms binding by full protein mass spectrometry. $n = 1$. **g** Confocal images of HEK293T expressing HTP-TM-SNAP. SNAP labelled with BG-JF$_{646}$; NBD-HTL stains cells unspecifically while NBD-SHTL labels extracellular HTP. **h** As for (**g**) but transfection with SNAP-TM-HTP. Imaging was performed for all conditions in N = 3 preparations with $n = 3$ images per condition. For all images, scale bar = 40 μm.

and HTP-TM-SNAP constructs, and indeed, observed surface labelling of NBD-SHTL when the HTP is extracellularly exposed (Fig. 3g) and intracellular labelling for NBD-HTL (Fig. 3h). As such, we have added a color in the green spectrum and demonstrate that the SHTL principle works on amide and amine HTL fused substrates.

## SiR-d12-SHTL reveals differential surface and internal mGluR2 localization in neurons

To test the SHTL approach in a biologically more complex system, we next used lentiviral particles to transduce primary mouse hippocampal neurons with a HTP-fused metabotropic glutamate receptor 2 (HTP-mGluR2) (Fig. 4a). This construct has previously been shown to maintain identical glutamate sensitivity[23] and trafficking[13] compared to the untagged receptor. mGluR2 is a family C GPCR involved in modulation of synaptic transmission and serves as a potential target for the treatment of neurological and psychiatric disorders[24]. Endogenously, mGluR2 is enriched along axons and presynaptic sites[25], although the precise sub-synaptic localization is not fully understood[24,26]. We aimed to determine the subcellular HTP-mGluR2 distribution by antibody staining against known markers of dendritic (MAP2), presynaptic (Bassoon) and postsynaptic (Shank2) sites (Fig. 4b). One week after transduction, we applied 500 nM of SiR-d12 (-HTL or -SHTL) to live cells for 30 minutes before

fixation and imaging on a confocal microscope (denoted as HTP(:SiR-d12)-mGluR2 and HTP(:S-SiR-d12)-mGluR2 for HTL and SHTL treatment, respectively). The HTL variant gave rise to pronounced signals stemming from intracellular sites in the soma (Fig. 4c, and Supplementary Fig. 9), quantified by defining regions of interest around cell bodies, localized by terminating dendritic MAP2 staining, and calculating the mean intensity. Non-transduced neurons did not show SiR labelling (Supplementary Fig. 9), indicating that SiR-d12-HTL selectively labels HTP-mGluR2 in all cellular compartments. By contrast, with SiR-d12-SHTL, the fluorescent signal was significantly reduced in the soma (Fig. 4d), with all remaining fluorescence appearing to stem from the cell surface, and in processes that were either MAP2 positive (Fig. 4e) or negative (Fig. 4f), revealing that a substantial intracellular population of HTP-mGluR2 likely exists in all 3 sites.

As an important control, anti-MAP2 intensities were identical in SiR-d12-HTL and SiR-d12-SHTL samples (Fig. 4g). Of note, MAP2 is known to locate in dendrites and not in axons, and most synapses in this preparation are axo-dendritic. To determine whether mGluR2 can be found in synapses, we next monitored the signal arising from apposing Bassoon and Shank2 puncta, and performed line scans (Fig. 4b, and Supplementary Fig. 9). Merging the channels (Fig. 4h) to perform line scans revealed a tendency towards Bassoon-positive pre-

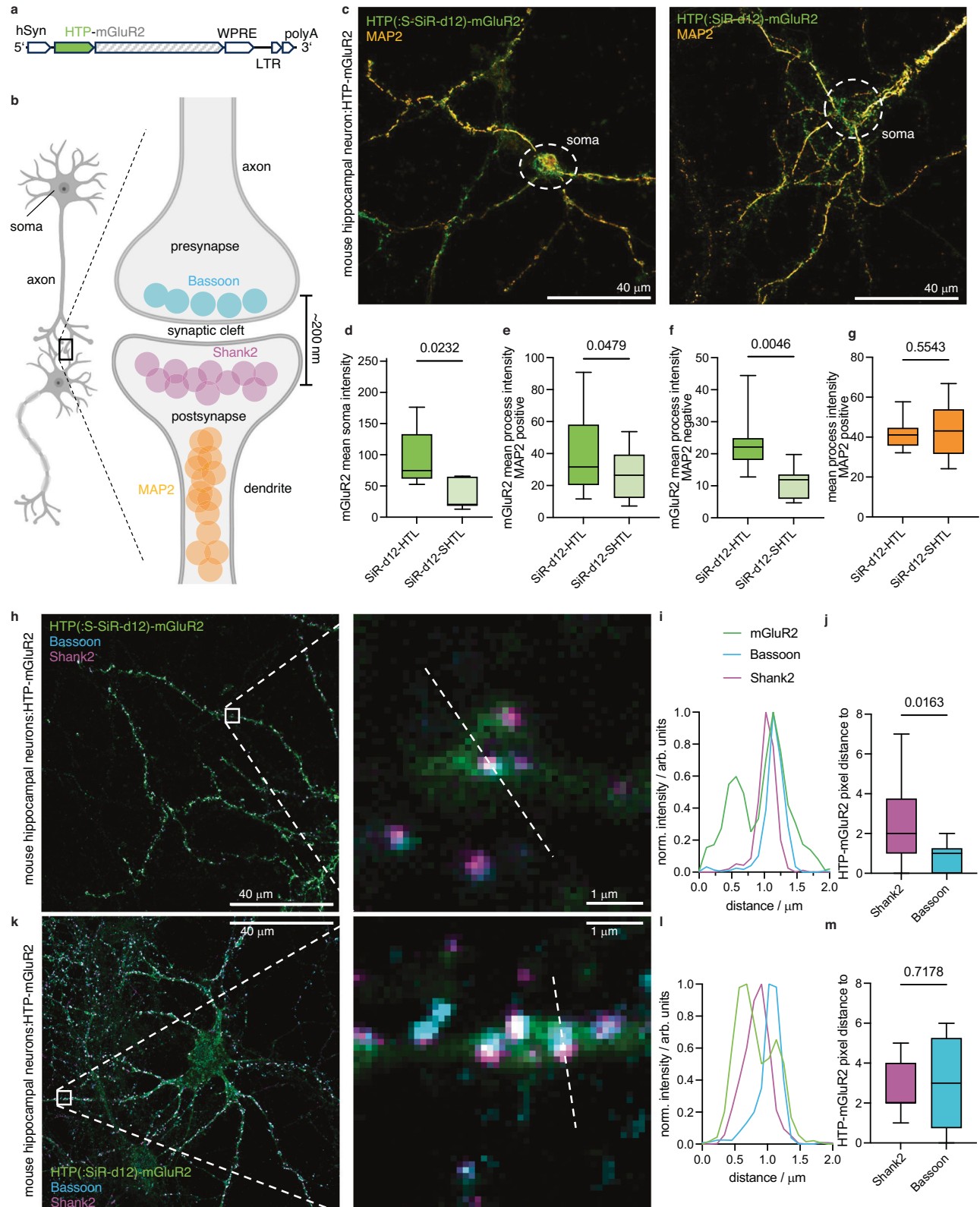

synaptic localization of HTP-mGluR2 (Fig. 4i), as revealed by plotting pixel distances of the respective intensity maxima (Fig. 4j). Such experiments outperform SiR-d12-HTL treated preparations (Fig. 4k), where signals likely include receptors in intracellular sites (Fig. 4l), therefore rendering synaptic allocation less precise, i.e., non-significant when comparing pixel distance (Fig. 4m). This underlines the critical impact of true surface labelling to interrogate glutamate-

responsive protein pools, since distances to release sites have been shown to be important for neural activity[27].

## Super-resolution imaging of HTP-mGluR2 at neural processes using SiR-d12-SHTL

After using confocal microscopy, we next turned to stimulated emission by depletion (STED) super-resolution imaging on these samples

**Fig. 4 | Revealing HTP-mGluR2 localization in hippocampal neurons. a** Viral DNA expression cassette with HTP-mGluR under hSyn promoter. **b** Neural connection via synapses and localization of dendritic MAP2, presynaptic Bassoon and postsynaptic Shank proteins (partly created in BioRender. Broichhagen, J. (https://BioRender.com/l5x8pjc.) **c** Confocal imaging of HTP-mGluR2 transduced mouse hippocampal neurons with SiR-d12-SHTL (left) and SiR-d12-HTL (right), co-stained with an antibody against MAP2 for dendrite identification. Scale bar = 40 µm. **d–g** Quantification of HTP-mGluR2 labelling in the soma (**d**), in dendrites (MAP2 positive, **e**) and in axons (MAP2 negative, **f**) reveals significantly less signal using SiR-d12-SHTL, while no difference in axonal MAP2 intensity is observed (**g**). Min-to-max box and whisker with median, SiR-d12-HTL: $n$ = 5 somata; mean = 92.9; min-max = 52.6–176.2 and SiR-d12-SHTL: $n$ = 8 somata; mean = 36.9; min-max = 13.0-65.7 for d, SiR-d12-HTL: $n$ = 14 processes; mean = 41.0; min-max = 11.6–90.8 and SiR-d12-SHTL: $n$ = 20 processes; mean = 26.5; min-max = 7.1–53.7 for e; SiR-d12-HTL: $n$ = 8 processes; mean = 23.7; min-max = 12.8–44.4 and SiR-d12-SHTL: $n$ = 8 processes; mean = 11.1; min-max = 4.7-19.8 for f, SiR-d12-HTL: $n$ = 14 processes, mean = 41.7; min-max = 32.2–57.7 and SiR-d12-SHTL: $n$ = 20 processes; mean = 43.9; min-max =

24.2-59.6 for g. 25-75% percentile for all. Unpaired Student's t-test two-tailed, normal distribution assumption for all. $p$ = 0.0232 (**d**), $p$ = 0.0479 (**e**), $p$ = 0.0046 (**f**) and $p$ = 0.5543 (**g**). **h** Confocal imaging of HTP-mGluR2 transduced mouse hippocampal neurons cells with SiR-d12-SHTL (500 nM), and the pre- and postsynaptic markers Bassoon and Shank2 (scale bar = 40 µm) with zoom-in (scale bar = 1 µm). **i** Representative line scan of a synapse shows mGluR2 co-localization primarily with the presynaptic marker Bassoon. **j** Quantification of mGluR2 localization with respect to Bassoon and Shank2. Min-to-max box and whisker with median, Shank2: $n$ = 10 synapses; mean = 2.60; min-max = 0-7 and Bassoon: $n$ = 10 synapses; mean = 0.80; min-max = 0–2. 25–75% percentile for both. Unpaired Student's t-test, two-tailed, normal distribution assumption. $p$ = 0.0163. **k–m** As for **h–j** but labelling with SiR-d12-HTL (scale bar = 40 µm) with zoom-in (scale bar = 1 µm). Min-to-max box and whisker, Shank2: $n$ = 10 synapses; mean = 2.70; min-max = 1–5 and Bassoon: $n$ = 10 synapses; mean = 3.00; min-max = 0–6. 25–75% percentile for both. Unpaired Student's t-test, two-tailed, normal distribution assumption. $p$ = 0.7178. Imaging was performed for all conditions from N = 3 preparations.

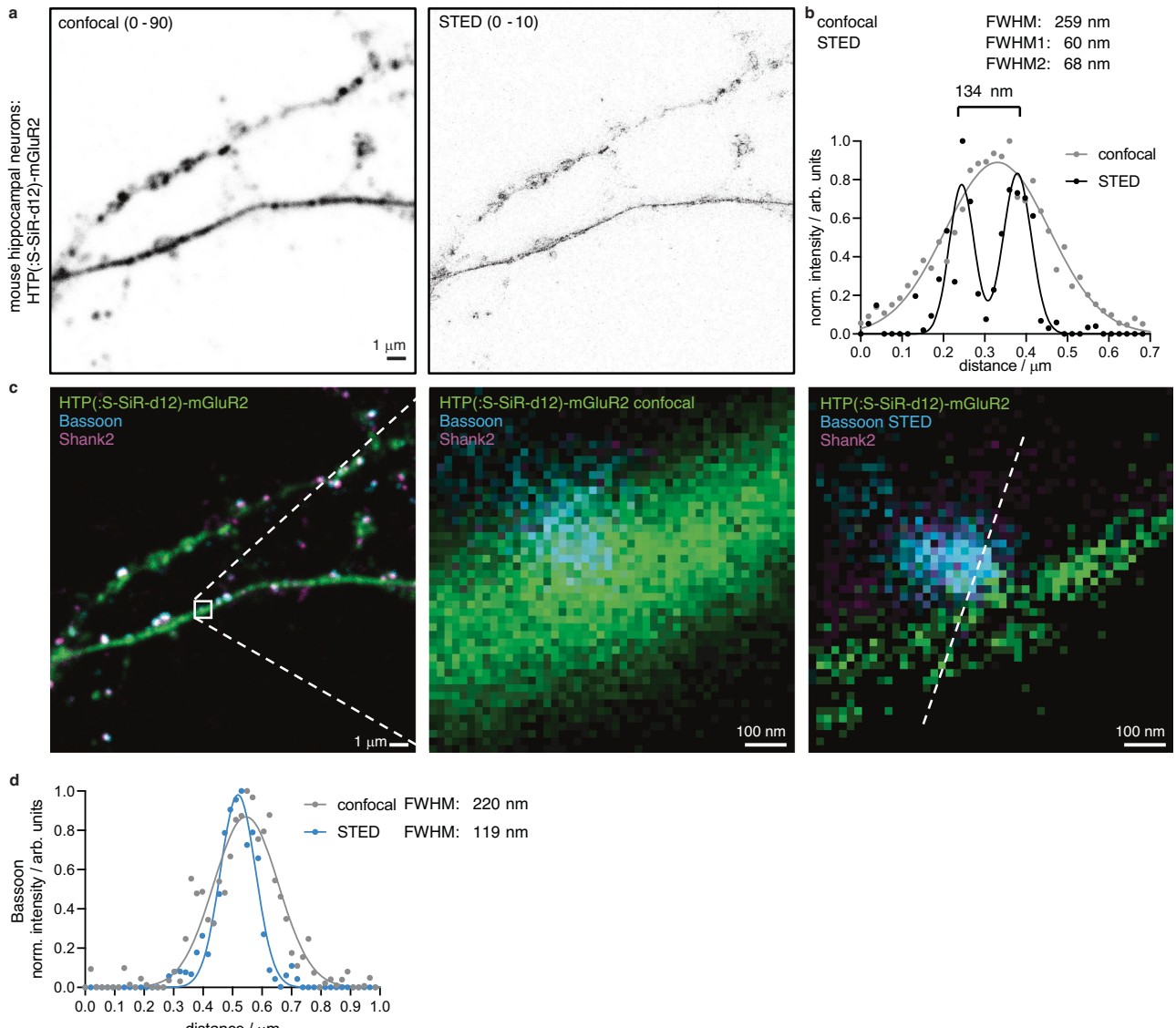

**Fig. 5 | Two-color STED super-resolution imaging of surface HTP(:S-SiR-d12)-mGluR2. a** Confocal (left) and STED (right) images of HTP(:S-SiR-d12)-mGluR2 transduced neurons. Scale bar = 1 µm. **b** Line scan profile of a process comparing confocal to STED performance, yielding a resolution of 134 nm across the ultrastructure. **c** Three color confocal and dual color STED with zoom in of the process reported in (**b**), (scale bar = 1 µm) with zoom-in (scale bar = 100 nm). **d** Line scan profile of the pre-synaptic marker Bassoon comparing confocal to STED performance allows the reporting of immunostained proteins additional to SHTL labelling. Imaging was performed from N = 3 preparations.

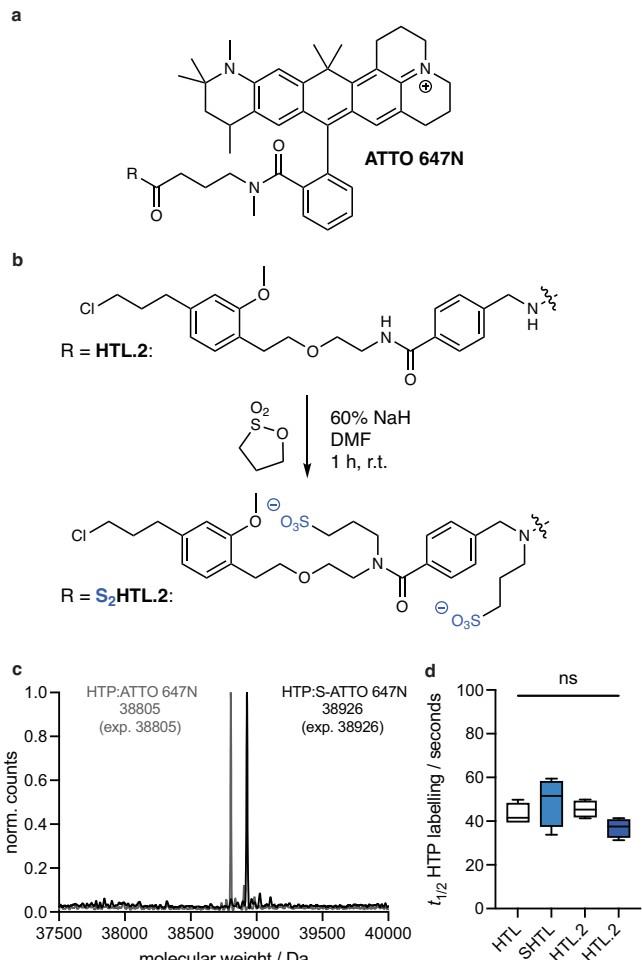

**Fig. 6 | Sulfonation on the HaloTag ligand v2.0 (HTL.2) with sticky ATTO 647N.** **a** Chemical structure of ATTO 647N, with a *N*-methyl amidated additional four carbon linker on the 3 position, which disallows proper dye:HTP secondary interactions. **b** Sulfonation protocol on second version HaloTag ligand HTL.2 yields double sulfonated $S_2$HTL.2. **c** qTOF full protein mass spectrometry of recombinant HTP labelled with ATTO 647N-HTL.2 and ATTO 647N-$S_2$HTL.2. $n = 1$. **d** Labelling kinetics of apo-HTP with 647N-HTL, -SHTL, -HTL.2 and $S_2$HTL.2. Min-to-max box and whisker with median, $n = 4$ for all conditions. HTL: mean = 43.1 s; min-max = 39.5–49.8 s, SHTL: mean = 49.1 s; min-max = 33.8–59.5 s, HTL.2: mean = 45.5 s; min-max = 41.3–49.9 s, $S_2$HTL.2: mean = 36.9 s; min-max = 31.3–41.4 s. 25-75% percentile for all. Ordinary one-Way ANOVA with normal (Gaussian) assumption. *ns* non-significant.

(Fig. 5a), as SiR-d12 is an excellent dye for this imaging technique[18]. Probing neuronal processes by a line scan revealed the localization of extracellular HTP-mGluR2 with a resolution of 134 nm (Fig. 5b) with full width at half maxima (FWHM) values of 60 and 68 nm (cf. confocal: FWHM = 259 nm). Furthermore, having a red fluorophore linked antibody targeted by immunocytochemistry against Bassoon, we were able to perform two-color STED on the contact site of a presynaptic site onto the process (Fig. 5c). Gaussian fitting gave a FWHM = 119 nm (cf. confocal: FWHM = 220 nm) (Fig. 5d), and since secondary antibodies are interchangable in color, this opens the door for more sophisticated imaging on HTP- and immunostained samples. This demonstrated the usefulness of addressing distinct pools of proteins in combination with different sets of proteins in proximity by super-resolution imaging, gaining information of (neural) ultrastructures.

## Taming ATTO 647N's stickiness using SHTL
ATTO 647N is a prominent carbopyronine dye that has been employed in STED[28,29] and structured illumination microscopy (SIM)[30,31] super-

resolution imaging, and its HTL derivative has been used to stain neurons in *Drosophila*[32]. It is characterized to be a bright dye in the far-red, which is favorable, and adds a third dye in the rhodamine(-like) class to the SHTL spectrum. However, when examining its molecular structure, we noted a four carbon linker on the C3 position for conjugation (Fig. 6a), which could exacerbate its intrinsic stickiness[33]. Moreover, we suspected that this linker would slow binding to HTP, given that secondary interaction sites between the HTP surface and rhodamine dyes are known to underlie the rapid labelling speed of dye-HTL conjugates, whereas ligands that do not bear a properly positioned dye or lack a dye are known to bind HTP inefficiently[34]. This limitation has been recently overcome by the design of a second-generation HTL.2 (ref. 35), which dramatically improves the efficiency of the DART (drug acutely restricted by tethering) approach[36].

To examine whether our sulfonation approach could extend to HTL.2, we synthesized ATTO 647N-HTL and ATTO 647N-HTL.2, before subjecting both to the sulfonation protocol to yield ATTO 647N-SHTL and ATTO 647N-$S_2$HTL.2, the latter of which bears two sulfonates as both amides are alkylated (Fig. 6b). To characterize nonspecific stickiness, we incubated non-transfected HEK293T cells with 1000 nM of each compound, observing massive non-specific staining for ATTO 647N-HTL, which was reduced for ATTO 647N-SHTL and ATTO 647N-HTL.2, and lowest for ATTO 647N-$S_2$HTL.2 (Supplementary Fig. 10).

We confirmed that the ATTO 647N-(S)HTL and ATTO 647N-($S_2$) HTL.2 cleanly react with recombinant HTP by mass spectrometry (Fig. 6c). We also investigated labelling kinetics by incubating 50 nM solutions of the HTL ligands with an excess of recombinant HTP (200 nM) while tracing fluorescence polarization. While this did not reveal detectible differences (Fig. 6d), the assay lacked the sensitivity needed to examine the biologically relevant case, in which low-nanomolar HTL in free solution outnumbers immobilized HTP molecules. We thus turned to the application of interest, using HEK293T cells encoding HTP-SNAP-mGluR2, incubated with HTL ligands in a titration series (1, 10 and 100 nM) for only 10 minutes before fixation. We used sparse expression with low amounts of DNA (50 ng *vs*. 400 ng as in Fig. 2) to ensure that the labelling reaction does not deplete ligand in free solution, particularly at the low concentrations. We co-labelled with the cell-impermeable BG-SulfoJF₅₄₉ (1 μM) to visualize the ideal pattern of surface-only labelling (note: HTP and SNAP are both extracellular).

The non-sulfonated ATTO 647N-HTL exhibited minimal labelling at 1 to 10 nM, while 100 to 1000 nM yielded predominantly nonspecific intracellular labelling that overshadowed any surface labelling (Fig. 7a–c). We believe this represents nonspecific stickiness as it was indistinguishable to that seen in mock-transfected cells, lacking HTP. The sulfonated first-generation ligand, ATTO 647N-SHTL, substantially reduced this nonspecific intracellular labelling in mock cells (Fig. 7d–f), however surface-specific HTP labelling in transfected cells was inefficient, appearing incomplete even at 100 nM; thus, surface labelling could be seen, but with bothersome nonspecific labelling. By contrast, the sulfonated second-generation ATTO 647N-$S_2$HTL.2 achieved clear labelling when delivered at only 100 nM, revealing bright surface labelling with negligible background (Fig. 7g). Even with 1 nM, clean surface labelling was evident after adjusting brightness and contrast (Fig. 7h, i). The non-sulfonated ATTO 647N-HTL.2 also exhibited efficient labelling (Supplementary Fig. 10), however, nonspecific intracellular uptake was seen if applied at 1000 nM. Thus, by combining HTL.2 with sulfonation, ATTO 647N-$S_2$HTL.2 performed better than all other ligands tested.

## A one-step protocol to synthesize dye-SHTLs for in situ use
The power of chemical biology is often limited by reagent availability. While popular probes are commercially available, variants developed in an academic setting are not always readily available, particularly over longer periods or in larger quantities. We decided to tackle this

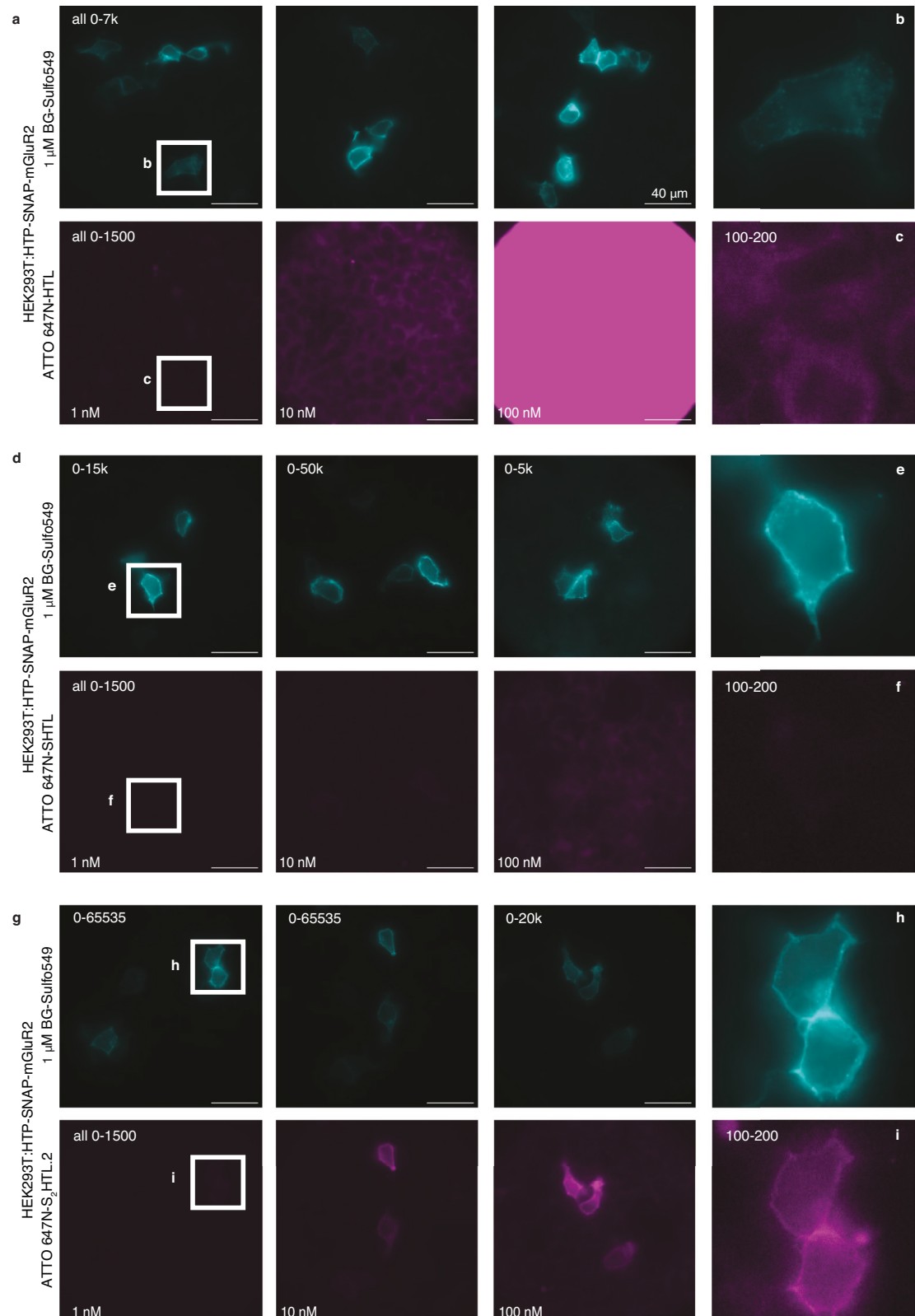

**Fig. 7 | Improved labelling with ATTO 647N-S₂HTL.2 using nanomolar concentration on sparsely mGluR2 expressing cells. a** HTP-SNAP-mGluR2 transfected HEK293T cells, labelled with BG-Sulfo549 (1 uM) and different concentrations of ATTO 647N-HTL.2 for 10 minutes prior to fixation and imaging gives rise to unspecific signal. **b, c** Zoom ins and brightness/contrast adjusted images from (**a**). **d**–**f** As for (**a**–**c**) but with different concentrations of ATTO 647N-SHTL leads to image improvements by removing unspecific signals. **g**–**i** As for (**a**–**c**) but with different concentrations of ATTO 647N-S₂HTL.2 allows clear membrane labelling even at 1 nM. Imaging was performed for all conditions in N = 3 preparations with *n* = 3 images per condition. For all images, scale bar = 40 µm.

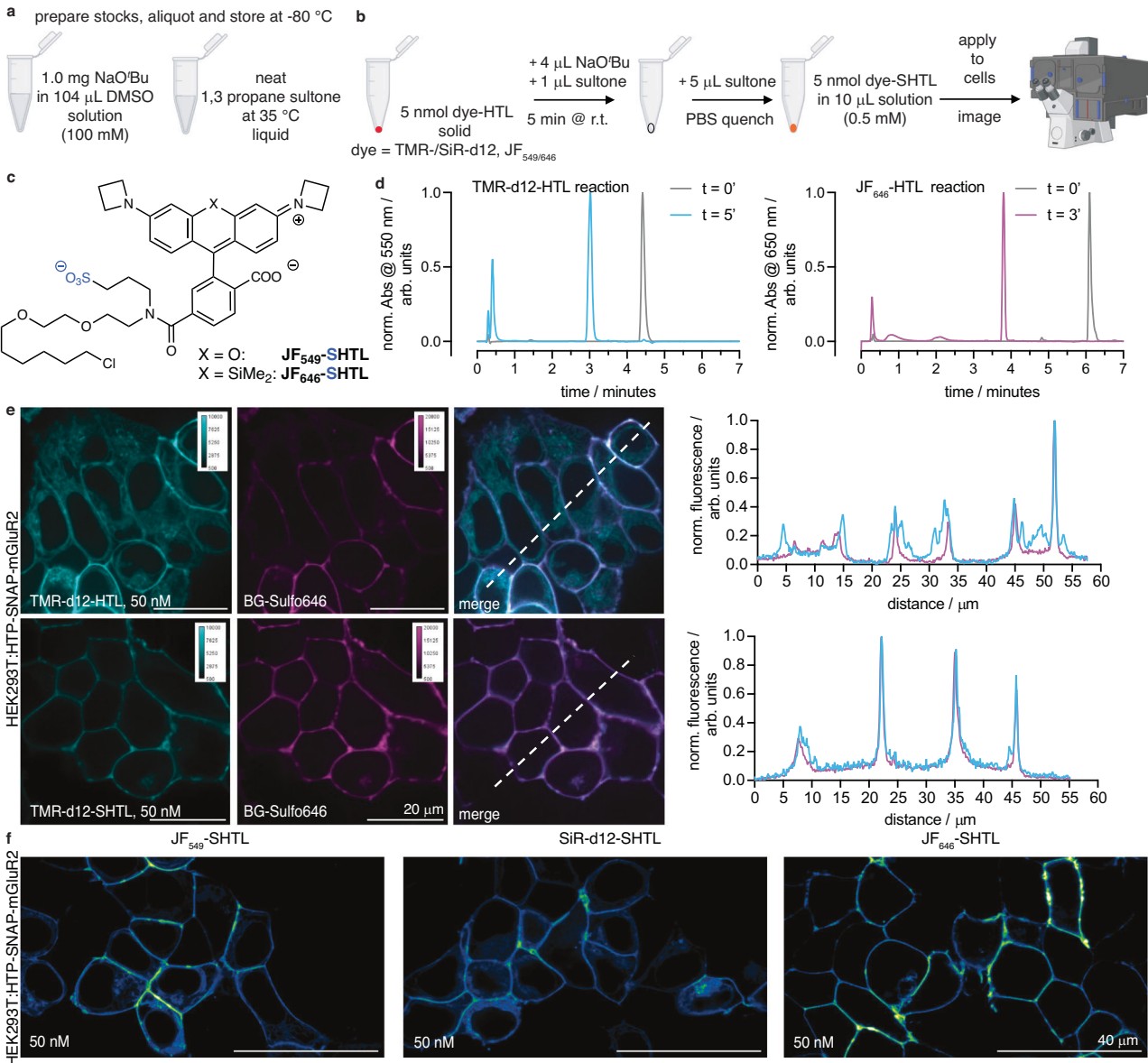

**Fig. 8 | One step protocol on small scale to synthesize and apply dye-SHTL.**
**a** Required stock solutions. **b** Outlined 5-minute synthetic protocol (partly created in BioRender. Broichhagen, J. (https://BioRender.com/l5x8pjc). **c** Structures of JF$_{549/646}$-SHTL. **d** LCMS traces of the reaction for TMR-d12-HTL prior reaction start and after 5 minutes and JF$_{646}$-HTL prior reaction start and after 5 minutes. $n = 1$. **e** Confocal imaging of HEK293T:SNAP-HTP-mGluR2 transfected cells with TMR-d12-(S)HTL (50 nM) and BG-Sulfo646 (50 nM) including line scans. Scale bar = 20 μm. **f** As for **e**, but with JF$_{549}$-SHTL, SiR-d12-SHTL and JF$_{646}$-SHTL. Scale bar = 40 μm. Imaging was performed for all conditions in N = 2 preparations with $n = 3$ images per condition.

issue by testing if our one-step protocol to make dye-HTL reagents impermeable can be performed on a small scale without the need for purification, so that any laboratory can perform the synthesis on demand. To facilitate this, we switched the base to sodium *tert*-butoxide, which is soluble to 100 mM in DMSO, making it easier to handle than sodium hydride (Fig. 8a). Neat 1,3-propane sultone was warmed to 35 °C (melting point 32 °C) to obtain a liquid that can straightforwardly be pipetted (Fig. 8a). We tested a quick protocol, for which 5 nmol of a dye-HTL is dissolved in 4 μL of base solution, before addition of 1 μL of the sultone. Incubation for 5 minutes at room temperature before quenching with 5 μL PBS yields a 0.5 mM dye-SHTL solution that can be directly used for cell application and imaging (Fig. 8b). We tested this protocol on commercially available HTL conjugates of JF$_{549}$ and JF$_{646}$, as well as HTL conjugates of our in-house developed probes TMR-d12 and SiR-d12 (Fig. 8c). Reactions were >99% quantitative according to LCMS analysis (Fig. 8d). It should be noted that the PBS quenching step

hydrolyzes the remaining sultone to an impermeable sulfonate, and neutralizes the base. Importantly, we observed no toxicity within 30 minutes of the reaction cocktail when incubating HEK293T cells with a 1:1000-dilution (resulting in: [DMSO] = 564 μM or 0.1%; [sultone] = 108 μM; [KO*t*Bu] = 4 μM) tested with a propidium iodide assay for cell death (Supplementary Fig. 11). Furthermore, after exposing HEK293T cells to this cocktail for 24 h, no effect on cell viability was observed using a WST-1 assay that measures metabolic impact (Supplementary Fig. 12). Higher concentrations led to cell death (LD$_{50}$ = 2.8 mM DMSO / 535 μM sultone / 20 μM KO*t*Bu), presumably caused by *tert*-butanol (i.p. LD$_{50}$ (mouse) = 399 mg/kg), supported by the fact that 14.1 mM DMSO concentrations did not affect viability, and no toxicity reports exist for 3-hydroxypropane-1-sulfonic acid in the National Library of Medicine. Therefore, we used a 1:10,000 dilution of the freshly prepared TMR-d12-SHTL (i.e., 50 nM) together with BG-Sulfo646 on HEK293T:SNAP-HTP-mGluR2 expressing cells (expressing

both tags on the extracellular side fused to metabotropic glutamate receptor 2) (Fig. 8e), and observed clear co-localization of both channels in confocal imaging. Similar performance was observed for JF$_{549}$-SHTL, SiR-d12-SHTL, and JF$_{646}$-SHTL (Fig. 8f), with the latter showing the cleanest performance.

## Discussion

The development of bright (and impermeable) fluorophores for microscopy has garnered increasing interest[11,37,38], particularly in the area of shadow imaging[39,40] and protein labelling[7,13,41,42]. In this study, we aimed to render existing and commercially available dye-HTL substrates impermeable to a cell's plasma membrane for a targeted approach, similar to strategies that have been applied to coumarin-arachidonic acid conjugates for uncaging a signalling lipid[43], BODIPY impermeabilization to tune cellular compartmentalization[44], Heidelberg Dyes for turn-on tetrazine click chemistry[45] and creating an impermeable version for the fluorescence-activating and absorption-shifting tag (FAST)[46]. In our case, we applied a quick and straightforward one step protocol on available rhodamine scaffolds, since these dye-HTL molecules only bear one acidic proton, and furthermore, in basic solutions form a non-fluorescent spirolactone form, preventing off-target alkylation on the carboxylate. This has been achieved here for the HaloTag system and distinguishes itself from other approaches, most importantly from equipping the dye with a charged group, which needs de novo synthesis, adding additional synthetic steps prior to fusing the dye with an SLP ligand (e.g., for rhodamine 6 G (5 steps)[47]; for JF$_{635}$i and Sulfo549/646 (3 and 5 steps, respectively)[6,7]). It is also in stark contrast to using commercially available impermeable dyes (e.g., Alexa Fluor 594, ATTO 532, SulfoCy3/5) and linking them to HTLs, since this would require coupling and HPLC purification, which we circumvent here with an in situ protocol. We obtained cis- and trans-amide mixtures (as per $^1$H NMR, see SI), and observed full protein labelling by mass spectrometry using recombinant HTP, without any significant difference in labelling kinetics via fluorescence polarization compared to non-sulfonated HTL substrates. The majority of dye-HTL reagents should be amenable to this method, however, the approach would not be applicable to certain scaffolds with other reactive attachment sites, including biotin, dyes with nucleophilic sites, for instance NH-anilines (e.g., ATTO 532, rhodamine 6 G, SiR595) or hydroxy groups (e.g., Oregon Green) as it would abolish their pharmacology (e.g., biotin will not bind streptavidin) or change the fluorescent properties (e.g., red-shift for ATTO 532, rhodamine 6 G, SiR595; extinguished fluorescence for O-alkylated Oregon Green). With such dyes not being attractive for our method, we chose nitrobenzodiazaisooxazole (NBD) to expand the repertoire of SHTL ligands. This provides further generalizability of sulfonation, adding a dye to the red/far-red color palette by showing fluorescence in the green spectrum, and secondly, broadening the protocol to other chemical moieties by addressing an amine (and not an amide) as the sulfonation site. Importantly, we demonstrate cellular impermeability of TMR-d12-, SiR-d12- and NBD-SHTL conjugates in live cell staining using widefield and confocal microscopy on HEK293T cells transfected with extra- and intracellular localized SNAP and HTP. Translating these finding to physiologically more relevant systems, we performed live-cell labelling of HTP-mGluR2 transduced hippocampal neurons with SiR-d12-SHTL prior to fixation, antibody staining and confocal imaging. By either specifically targeting the extracellular, exposed HTP-mGluR2 protein pools with SiR-d12-SHTL, or the total protein pool with SiR-d12-HTL, we found that there is a mix of surface and intracellular populations in the soma and both dendritic and axonal processes. Critically, SiR-d12-SHTL allowed us to find the presence of a surface pool of axonal and presynaptically-localized mGluR2. As the trafficking mechanisms and nanolocalization of mGluRs is a highly active field of study[24,26,27,48], our observations point towards many surface

and intracellular subpopulations of mGluR2, both in the soma and processes/synapses, which will warrant future study, amenable to super-resolution STED nanoscopy. To this end, this represents state-of-the-art neuronal imaging, as STED has recently helped to understand nanoarchitecture of synapses in general[49,50], and in particular of disease-relevant proteins, such as Munc13-1 (involved in the exocytotic machinery)[51], on SYP1/SYT1/VGLUT (markers for synaptic vesicles)[52], Tenm3-Lphn3 complexes (that reconstitute synaptic junctions)[53], Blobby (an active zone assembly protein necessary for memory formation in *D. melanogaster*)[54] and NMDA receptors (in a trans-synaptic context)[55]. Although we used an overexpression system, our findings open up interesting avenues to probe GPCRs and other receptors in different cellular compartments with precision for surface versus intracellular pools, which we aim to perform on endogenous GPCRs in the central nervous system, similar as to our reported Glp1r$^{SNAP/SNAP}$ animals to demonstrate receptor localization in the brain and using super-resolution imaging[42]. This is needed for a full physiological understanding, as overexpression may lead to aberrant dimer or oligomer formation of GPCRs[56,57], alter downstream signaling properties (e.g., ERK activation[58], pro-survival[59], counteracting metabolic disfunction[60]) and localization[61]. For these reasons, we showcase ATTO 647N on the HTL.2 substrate to exhibit cleaner labelling even at low doses down to 1 nM, on cells with vastly lower expression levels, and an incubation time of only 10 minutes. These parameters may be essential when applied to complex tissues or in live animals, where high concentrations are difficult to achieve, endogenous proteins are ideally tagged, and short incubation times come in handy. In addition, ATTO 647 N has been reported to be sticky, which may lead to nonspecific background labeling[62,63], and thus the addition of two negative charges on S$_2$HTL.2 offers even greater attenuation of nonspecific stickiness. On top of this, employing a carbopyronine scaffold, we added another dye class into SHTL's payload repertoire. Other dyes that may be interesting, especially due to their performance in the near-infrared, include 2XR-1 (ref. 64) and heptamethine cyanines[65]. Finally and generally, a one-step protocol with no need for purification that produces compounds for direct application are attractive in biological sciences[66,67], and we have added to this portfolio. Although the reaction is near-quantitative, small amounts of starting material left may give rise to intracellular staining (Supplementary Fig. 13). As such, the use of far-red JF$_{646}$-SHTL alongside with 10,000-fold dilutions of the reaction mixture are recommended for surface labelling. We provide a one-page step-by-step protocol for straightforward implementation in the Supplementary Information.

In summary, we report the synthesis and application of sulfonated HaloTag substrates, which can be performed in one step on amines and amides, to label HTP-fused cell surface proteins in the green, red, and far-red color spectrum. Validated in live cell widefield and confocal imaging in transfected HEK293T cells, we demonstrate confocal and dual-color STED super-resolution microscopy on HTP-mGluR2 transduced hippocampal neurons, confirming the impermeability of the newly-synthesized dyes in more complex biological systems with the ability to precisely localize the surface population of a GPCR involved in neurotransmission. We expand the strategy to the second-generation HTL.2, and provide a simple, 10-minute sulfonation protocol that any laboratory can execute on miniature scale−on commercially available JF-HTLs−without the need for purification for direct application. We envision wide applicability.

## Methods

### Ethical Statement

Experiments utilizing mice were performed under the ethical regulation of the European law and the state of Berlin (Landesamt für Gesundheit und Soziales; LAGeSo). Mice (*Mus musculus*, C57BL/6 J background, purchased from Charles River) were kept at the animal

house of the Leibniz-Forschungsinstitut für Molekulare Pharmakologie under IVC conditions, on a 12 h light, 12 h dark cycle, at room temperature (20 ± 1 °C) and humidity levels of ~50%, with access to food and water ad libitum. Mice were bred and postnatal day 0–1 offspring of both sexes were used to generate neuronal cultures for imaging (Figs. 4–5). Each culture was obtained from one or more mice, and three independent cultures were used for imaging. The sex of the animals was not considered as a variable in the study, and a sex-based analysis was not performed in this study.

## General

All chemical reagents and anhydrous solvents for synthesis were purchased from commercial suppliers (Sigma-Aldrich, Roth) and were used without further purification if not stated otherwise. The synthesized and used compounds meet the community requirements as per ref. 68.

NMR spectra were recorded at 300 K in deuterated solvents on a Bruker AV-III spectrometer using a room- temperature 5 mm broadband probe equipped with one-axis self-shielded gradients and calibrated to residual solvent peaks ($^1$H/$^{13}$C in ppm): CDCl$_3$ (7.26/77.0), DMSO-d$_6$ (2.50/N.A.), MeOD-d4 (3.35/N.A.), D$_2$O (4.65/N.A.). Multiplicities are abbreviated as follows: s = singlet, d = doublet, t = triplet, q = quartet, p = pentet, h = heptet, br = broad, m = multiplet. Coupling constants $J$ are reported in Hz. Spectra are reported based on appearance, not on theoretical multiplicities derived from structural information.

UPLC-UV/Vis for purity assessment was performed on an Agilent 1260 Infinity II LC System equipped with Agilent SB-C18 column (1.8 μm, 2.1 × 50 mm). Buffer A: 0.1% FA in H$_2$O Buffer B: 0.1% FA acetonitrile. The typical gradient was from 10% B for 1.0 min → gradient to 95% B over 5 min → 95% B for 1.0 min with 0.6 mL/min flow or from 30% B for 1.0 min → gradient to 95% B over 5 min. For ATTO 647N-HTL, ATTO 647N-SHTL and ATTO 647-S$_2$HTL.2 50% B for 1.0 min → gradient to 95% B over 5 min was used and for ATTO 647-HTL.2 70% B for 1.0 min → gradient to 95% B over 5 min was used instead. Retention times ($t_R$) are given in minutes (min). Chromatograms were imported into GraphPad Prism 10 and purity was determined by calculating AUC ratios.

Preparative or semi-preparative HPLC was performed on an Agilent 1260 Infinity II LC System equipped with columns as followed: preparative column –Reprospher 100 C18 columns (10 μm: 50 x 30 mm at 20 mL/min flow rate; semi-preparative column – 5 μm: 250 x 10 mm at 4 mL/min flow rate. Eluents A (0.1% TFA in H$_2$O) and B (0.1% TFA in MeCN) were applied as a linear gradient. Peak detection was performed at maximal absorbance wavelength.

For small molecule HRMS, samples were analyzed on Orbitrap Fusion mass spectrometer (Thermo Fisher Scientific). MS scans were acquired in a range of 350 to 1500 m/z. MS1 scans were acquired in the Orbitrap with a mass resolution of 120,000 with an AGC target value of 4e5 and 50 ms injection time. MS2 scans were acquired in the ion trap with an AGC target value of 1e4 and 35 ms injection time. Precursor ions with charge states 2-4 were isolated with an isolation window of 1.6 m/z and 40 sec dynamic exclusion. Precursor ions were fragmented using higher-energy collisional dissociation (HCD) with 30% normalized collision energy. Samples were measured $n = 1$. Small molecule samples were prepared as 10 μM solutions in JB Special medium (H$_2$O/MeCN/HOAc = 25/25/1).

Full protein MS was measured using a Waters XEVO G2-XS quadrupole time-of-flight (Q-TOF) mass spectrometer (Waters cooperation, USA) alongside an, Acquity UPLC system with an Acquity UPLC Protein BEH C4 column (1.7 μm, 2.1 mm x 50 mm). Samples were eluted with a flow rate of 0.4 mL/min. The following gradient was used: A: 0.01 % FA in H$_2$O; B: 0.01 % FA in MeCN. 5-85 % B: 0–3.5 min; 85-5 % B: 3.7-4 min; 5-85 % B: 4-4.5 min; 85-5 % B: 4.5-5.5 min. MS was acquired over the range of 500-4000 Da using continuous scanning with 1 s scan time and a cone voltage of 80 V with subsequent spectral deconvolution

using the MaxEnt1 algorithm at a resolution of 1 Da/channel until convergence. Samples were measured $n = 1$. All samples were prepared as 0.1 mg/mL solutions in PBS.

## Molecular modelling

A detailed description for Molecular Modelling can be found in the Supplementary Information. The isolated ligand geometries were pre-optimized using OpenBabel 3.1 (MMFF94, conjugated gradient algorithm, 3D conformer generation setting „best")[69–74]. Subsequently, the structures were geometry optimized using the ORCA 6.1 quantum chemistry suite[15]. Here, the ωB97X-D3BJ/def2-TZVPP[SMD(H$_2$O)] method was used with the default integration grid settings and both tight SCF and tight optimization convergence criteria. The RIJCOSX system was employed throughout with automatic auxiliary basis set selection[75,76]. After geometry convergence, the stability of the minimum was validated by calculating the vibrational frequencies using the same method.

The receptor geometry was obtained from PDB-5UY1[17]. The receptor was prepared for docking using standard setup protocols[77,78]. All docking experiments were performed using the GNINA 1.1 docking program, whereas the empirical scoring function „Vinardo" was employed in tandem with the CNN model „general_default2018", the latter was used for re-scoring of the final poses[16,79]. All experiments used an exhaustiveness of 128, whereas a total of 50 output poses was requested. The random seed was set to 42. The search box was defined using the full receptor geometry, whereas the autobox method was used to fully enclose the receptor including 5.0 Å padding on all sides, and allowing for automatic box extension.

## Cell culture, transfection and staining

HEK293T cells (German Collection of Microorganism and Cell Cultures) were cultured in growth media (DMEM, Glutamax, 4.5 g Glucose, 10% FCS, 1% PS; Invitrogen) at 37 °C and 5% CO$_2$. 50 000 cells per well were seeded on 8-well μ slides (Ibidi) previously coated with poly-L-lysine (Aldrich, mol wt 70 000–150 000). The next day, 400 ng DNA was transfected using 0.8 μL Jet Prime reagent in 40 μL Jet Prime buffer (VWR) per well/plasmid. Media was exchanged against antibiotic-free media before the transfection mix was pipetted on the cells. After 4 hours incubation at 37 °C and 5% CO$_2$, medium was exchanged against growth media, and after an additional 24 hours, cells were stained. All dyes for widefield imaging were used at a concentration of 500 nM with the addition of Hoechst 33342 at 1 μM, in growth media. Cells were stained at 37 °C and 5% CO$_2$ for 30 minutes. Afterwards cells were washed once in growth media and imaged in fluorobrite (Invitrogen). For confocal microscopy all dyes were additionally used at a concentration of 50 nM.

## Live cell widefield imaging

Living cells were imaged in fluorobrite (Invitrogen) using an epi-fluorescence Nikon Ti-E microscope, equipped with pE4000 (cool LED), Penta Cube (AHF 66-615), 60× oil NA 1.49 (Apo TIRF Nikon) and imaged on a sCMOS camera (Prime 95B, Photometrics) operated by NIS Elements (Nikon). For excitation the following LED wavelengths were used: Hoechst – 405 nm, JF$_{549}$, TMR-d12 – 550 nm, JF$_{646}$, SiR-d12, ATTO 647 N – 635 nm. Line scans were drawn with the polygon selection in ImageJ2 (version 2.14.0/1.54p) and the grey values obtained and plotted against the distance in Graphpad Prism 10, in which values were normalized

## Propidium iodide assay

HEK293T cells were cultured in growth media (DMEM, Glutamax, 4.5 g Glucose, 10% FCS, 1% PS; Thermo Fisher Scientific) at 37 °C and 5% CO$_2$. 50,000 cells per well were seeded on 8-well μL slides (Ibidi) previously coated with poly-L-lysine (Sigma Aldrich, mol wt 70,000–150,000). The next day cells were incubated with the DMSO/KOtBu/sultone

(1:1,000) in DMEM fluorobrite (Thermo Fisher Scientific) containing 0.25 μM propidium-iodide (Thermo Fisher scientific, P3566). As a positive control TritonX-100 1:250 of a 9 wt% solution in $H_2O$ was used. Cells were imaged over 30 min, 10 frames using a Ti-E Nikon epi-fluorescence microscope equipped with pE4000 (cool LED), Penta Cube (AHF 66-615), 60x oil NA 1.49 (Apo TIRF Nikon) and imaged on SCMOS camera (Prime 95B, Photometrics) operated by NIS Elements (Nikon). For excitation the following wavelength was used: propidium-iodide: λ = 550 nm.

## WST-1 Assay
HEK293 cells were seeded (12000 cells/well) in clear 96 well plates (Greiner) and incubated overnight in 100 μL DMEM supplemented with 10% FBS at 37 °C, 5% $CO_2$. The following day, the DMSO / KOʳBu / sultone cocktail (1:100; 1:500; 1:1,000; 1:5,000; 1:10,000; 1:50,000; 1:100,000 dilution series), 10 μM staurosporin or vehicle (volume corresponding to dye volume) were prepared in full media and then added in six replicates to the cells along with one well containing no cells and placed in the incubator. After 24-hour incubation WST-1 (#MK400, Takara Bio) was added to the cells according to the manufacturer's instructions, i.e., 1:10 dilution. Absorbance was read after 2.5 hours on a TECAN INFINITE M PLEX plate reader ($λ_{Abs} = 440$ nm) and corrected by subtraction ($λ_{Abs\ correct} = 660$ nm). The data was plotted using GraphPad Prism 10.

## FLIM
Drops of 10-20 ul of 2-5 μM Fluorophor where spotted in μ-Slide 8 Well Glass Bottom (Ibidi #80877). Fluorescence lifetime was measured on a Leica SP8 TCS STED FALCON (Leica Microsystems) equipped with a pulsed white-light excitation laser (80 MHz repetition rate, NKT Photonics), a 100× objective (HC PL APO CS2 100×/1.40 NA oil), a temperature-controlled chamber at room temperature, operated by LAS X. A Hybrid detector produces FLIM images of 512 × 512 pxl with 113 nm per pxl after 10 frame repetitions. Solution of 5 μM Fluorescein was used as a reference, excited at 488 nm and the lifetime of the collected em from λ = 503-580 nm was 3.8 ns. HTL and TMR-d12-SHTL solutions (2 μM) were excited using λ = 561 nm, emission signals were captured at λ = 577–650 nm before and after addition of 10 μM purified HaloTag protein. HTL and SiR-d12-SHTL were excited using λ = 640 nm, emission signals were captured at λ = 655–750 nm. Fluorescence lifetime decay curves from selected regions were fitted with one exponential function and the lifetime is reported for each region.

## Hippocampal neurons
Primary hippocampal neurons were cultured according to a protocol described in ref. 80: hippocampi were dissected from wild-type C57BL/6 J P0-P1 mice, and were incubated in 37 °C gently shaking in a solution containing 0.2 mg/ml L-cysteine, 1 mM $CaCl_2$, 0.5 mM ethylenediaminetetraacetic acid (EDTA) and 25 units/ml of papain (Worthington Biochemicals), pH 8 in Dulbecco's modified eagle medium. After one hour, the solution was replaced by a prewarmed solution containing 2.5 mg/ml Bovine Serum Albumin, 2.5 mg/ml trypsin inhibitor (Sigma-Aldrich T9253), 1% Fetal Bovine Serum, heat inactivated (FBS; e.g., Gibco™ A15-104) in Dulbecco's modified eagle medium, and the hippocampi were incubated at 37 °C gently shaking. The solution was then replaced by neuronal culture medium (Neurobasal™-A medium supplemented with 2% B-27™ Plus Supplement, 1% GlutaMAX™ supplement, and 1% Penicillin-Streptomycin), and the hippocampi are gently triturated to produce a cell suspension that was then plated on PLL-coated coverslips for 2–3 weeks at 37 °C and 5% $CO_2$ before experiments began. The culture contains primarily neurons and few astrocytes, and is thus ideal for imaging. At day-in-vitro 1-2, neurons were infected with lentiviral particles encoding for HTP-mGluR2, prepared by the viral core facility (Charité Universitätsmedizin Berlin) according to the protocol published in ref. 81 and modified as in ref. 82, using a modified version

of the Addgene plasmids #8454 and #8455 (ref. 83) and under the synapsin promoter. At day-in-vitro 14-17, neurons were incubated for 30 minutes with the indicated dyes at 37 °C and 5% $CO_2$, washed two times in PBS to remove unbound dyes and cell debris, fixed with cold 1% paraformaldehyde (PFA) in PBS for 10 minutes, and then washed two times in PBS to remove the PFA. The cells were permeabilized with cold 0.25% Triton-X-100 in PBS for 10 minutes and rinsed once in pure PBS. To block nonspecific binding sites, the cells were incubated in cold 0.3% NGS in PBS for 20 minutes. Next, neurons were stained with the following antibodies for 1 hour at RT: Guinea pig polyclonal Shank2 (Synaptic Systems 162 204, diluted 1:250), Chicken polyclonal MAP2 (Novus Biologicals NB300-213, diluted 1:1000), and Mouse monoclonal anti Bassoon (Abcam, AB_82958, clone SAP7F407, diluted 1:250), in PBS containing 0.3% NGS. To remove unbound antibodies, the cultures were washed three times with blocking solution before applying the following secondary antibodies conjugated with fluorophores: Anti-Chicken-AF405 (Abcam AB_175674), Anti-Guinea Pig-CF488 (Biotium CF® AB_20169-1) and Anti-Mouse-AF594 (Invitrogen A32744). The secondary antibodies were diluted 1:500. The neurons were incubated with the secondary antibodies for 45 minutes at RT in the dark and washed eight times in PBS to remove unbound products. Next, neurons were fixed with 4% PFA in PBS for 15 minutes at RT and then washed two times in PBS to remove excess PFA. The coverslips were rinsed once by dipping shortly into $ddH_2O$, mounted with ProLong™ Gold Antifade Mountant (Invitrogen™ P36934) on microscope glass slides, and stored in the dark at RT for 48 hours to dry before imaging.

## Confocal and STED imaging
Confocal microscopy on transduced neurons was performed using a Leica SP8 TCS STED FALCON (Leica Microsystems) equipped with a pulsed white-light excitation laser (80 MHz repetition rate, NKT Photonics), a 100x objective (HC PL APO CS2 100×/1.40 NA oil), a temperature-controlled chamber and operated by LAS X. SiR-d12 was excited using λ = 647 nm and emission signals were captured at λ = 656-751 nm. Confocal images were collected using a time gated Hybrid detector (0.5–6 ns). Images of 1024 x 1024 pixel had a pixel size of 113.64 nm. Time-gated detection for STED was set to the same values for all dyes and emission was detected within 560-643 and 650-751 nm for Bassoon antibody and SiR-d12, respectively, with sequential excitation. The 775 nm STED laser was used to deplete both fluorophores. Regions of interest were manually drawn using ImageJ2 (version 2.14.0/1.54p), and mean gray values plotted in Graphpad Prism 10. Statistics were calculated in Graphpad Prism 10.

## Quantum yield
Absolute quantum yields were measured via steady-state UV-Vis absorption on a Hamamatsu PL Quantum Yield Spectrometer C11347 equipped with an integrating sphere. The solutions were prepared to have an absorbance between 0.05 and 0.1 at 549 nm for TMR in PBS and at 652 nm for SiR in PBS. All measurements were done in 1 cm quartz cuvettes.

## Reporting summary
Further information on research design is available in the Nature Portfolio Reporting Summary linked to this article.

# Data availability
All experimental data, materials and methods, analytical procedures, cell assays and copies of spectra are available in the Article and its Supplementary Information. Source data are provided with this paper.

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

## Acknowledgements

This project has received funding from the European Union's Horizon Europe Framework Programme (deuterON, grant agreement no. 101042046 to JB). This work was supported by the German Research Foundation Excellence Strategy EXC-2049-390688087 (NL) and CRC 1286 "Quantitative Synaptology" project A11 (NL). HTL.2 reagents were supported by NIH grants 1RF1MH117055 and 1DP2MH1194025 (to MRT). JL is supported by NIH grant R01NS129904. We thank K. Steinhagen and animal facility staff at the Leibniz-Forschungsinstitut für Molekulare Pharmakologie for technical help. We thank the Viral Core Facility, Charité Universitätsmedizin Berlin, for virus production, and B. Weinberg for the contribution of lentiviral vectors (Addgene, #8454 and #8455).

## Author contributions

Conceptualization and Methodology: JB; Formal analysis and investigation: K.R., U.P., B.C.B., S.S., C.H., C.H.O., M.K., E.T., M.B., R.B., J.L., M.L., N.L. and J.B.; Provision of reagents: B.C.S., P.J., J.H., M.R.T. Writing–Original Draft: J.L. and J.B.; Reviewing and Editing: K.R., U.P., J.L., N.L., M.L., B.C.S., J.H., M.R.T. and J.B.; Visualization: K.R., U.P. and J.B.; Supervision: N.L. and J.B.; Funding Acquisition: J.L., N.L. and J.B.

## Funding

## Competing interests

NL is a member of the scientific advisory board of Trace Neuroscience. MRT and BCS are on patent applications describing HTL.2. All other authors declare no competing interests.
