## [Transparent Peer Review file · Nature Communications]

A general one-step protocol to generate impermeable fluorescent HaloTag substrates for in situ live cell application and super-resolution imaging

Corresponding Author: Dr Johannes Broichhagen

Version 0:

Reviewer comments:

Reviewer #1

(Remarks to the Author)

The authors present a practical and streamlined solution to a well-known challenge in surface protein labelling. Given its simplicity and adaptability, this method has clear potential for widespread use in cell and molecular biology.

The authors introduced a compelling method to differentiate between the surface and intracellular pools of HTP-tagged mGluR2. Nonetheless, considering that GPCR trafficking and subcellular localisation can be highly sensitive to expression levels, this limitation requires a more thorough examination. Given the application of this technique, any additional data or discussion on how overexpression affects receptor distribution is of interest.

Furthermore, the manuscript would benefit from a more complete and transparent description of the docking methodology. The authors mention the use of the “two-point attractor method” for covalent docking in AutoDock 4, which is a valid and increasingly adopted strategy originally described by Bianco et al. (2015). However, the implementation details are not fully explained in this manuscript. It would be helpful to briefly summarise the key features of the method and how it was adapted for this particular system, especially for readers who are unfamiliar with covalent docking workflows. In addition, critical parameters, such as the choice and justification of the docking grid size, the definition of the binding site, and how covalent bond formation was enforced during the simulations, were not sufficiently reported. The manuscript should also clarify whether the reported binding affinities correspond to AutoDock’s internal scoring functions. However, these scores do not represent the true physical binding free energies, which is an important limitation that should be acknowledged. Finally, the authors might briefly comment on whether alternative scoring approaches could be applied and whether different docking decoys could be expected.

Reviewer #2

(Remarks to the Author)

In this manuscript, the authors report a method they describe as “a general one-step protocol to generate impermeable fluorescent HaloTag substrates,” with demonstrations in super-resolution imaging. While the biological applications are well-executed and add value, the chemical aspect of the work does not present a significant advance over existing methods. The approach primarily involves incorporating a charged sulfonic acid group into the fluorophore-conjugated HaloTag ligand—a strategy that has already been explored in various forms, including placing the HaloTag ligand and the anionic group at different positions on the fluorophore.

The modification of rhodamine dyes with functional groups such as sulfonates is a well-established technique and can often be done without substantially altering the photophysical properties of the dyes. As such, the chemical novelty of this method appears limited, and its distinction from previously published approaches is unclear.

The authors conducted extensive biological experiments to validate their approach; however, the applications demonstrated are fairly typical for super-resolution fluorescence microscopy. Additionally, the deuterated rhodamine dyes (TMR-d12 and SiR-d12) show only marginal improvement over their non-deuterated counterparts and do not offer a substantial performance benefit for super-resolution imaging.

Therefore, I rejected this work for publication in Nature Communications.

Reviewer #3

(Remarks to the Author)

In this manuscript, the authors described a new strategy to make HaloTag ligand (HTL) impermeable to cell membrane via simple chemically synthetic method, so that HaloTag-conjugated proteins (HTP) can be labeled only on the cellular surface. They demonstrated selective surface protein labeling to live HEK293 cells and hippocampal neurons genetically modified with HTP. The authors also proposed a simple and efficient protocol for synthesizing cell-impermeable HTL from known HTLs. The experimental procedures seem technically sound and the results look appropriately processed and presented. The authors demonstrated synaptic mGluR2 localization and found the relatively diverse distribution of the receptors in detail, by means of their new sulfonated HTP (SHTP) and super resolution fluorescence microscopy. The also showed their strategy is applicable and useful to conventional confocal microscopy. Their approach for clear and selective labeling to membrane surface proteins would be successful and useful to track the proteins-of-interest, and the combination of a simple chemical modification on the HTL with genetical manipulation of the target cells/tissue is attractive in a view point of chemical biology. However, the basic technologies employed in this study would be well established except their molecular design and modification reaction to introduce sulfonate groups with sultone. Although the optimization of the position of medication and the reaction conditions might be not straightforward, it would be rather chemical issue how the utility of sulfonate groups increases with simple introduction procedures for many kinds of useful dyes. The method the authors proposed has still limitation to synthesize sulfonated ligands such as stability for strong base reagents, while there are already several strategies to develop sulfonated fluorescence dye parts as the authors mentioned, even though they also have their own limitations. The work is interesting and attractive, but rather suitable for the specific community of organic chemistry and chemical biology.

Version 1:

Reviewer comments:

Reviewer #1

(Remarks to the Author)

The authors have addressed all the methodological concerns raised in the previous revision round. The revised Supplementary Information now contains sufficient detail to ensure the full reproducibility of the computational workflow, including ligand optimization, docking configurations, re-scoring, and statistical analysis. The figures and tables have been significantly improved and now convincingly support the conclusions. The revision significantly strengthened the manuscript, and the claims are now well-supported.

Reviewer #2

(Remarks to the Author)

In the revised version of the manuscript, the authors have addressed most of the comments raised by the reviewers. The authors synthesized a series of modified dyes and conducted extensive biological experiments to demonstrate a one-step protocol for generating impermeable fluorescent HaloTag substrates. The novelty of the work primarily lies in modifying the molecular structure by introducing a sulfonate anionic group into the HTL moiety rather than incorporating the anionic group directly onto the dye scaffold, which indeed simplifies the synthetic process to some extent. The study is new and of interest.

However, this improvement does not constitute a substantial advance in the development of improved fluorescent dyes for in situ live-cell applications. Since HTL is covalently conjugated to the dye, it can essentially be regarded as part of the dye molecule. Thus, the distinction between introducing a sulfonate group onto the HTL moiety versus directly onto the dye is not particularly significant. Additionally, from an organic synthesis standpoint, incorporating a sulfonate group is not technically challenging. Although the authors performed numerous imaging experiments, including confocal and super-resolution microscopy, these data alone do not substantively enhance the novelty of the work.

Therefore, I agree with Referee 3 that this manuscript may be more suitable for a specialized journal in organic chemistry and chemical biology rather than a broader readership.

Reviewer #3

(Remarks to the Author)

In this revised manuscript, the authors politely responded to the comment from this reviewer although the comment from this reviewer contains ambiguous points. Reviewing the additional experimental results such as new Figure 3, this reviewer understands that the strategy of their work is applicable to another dye, NBD-HTL, via same synthetic methodology, and also that this strategy is easy to adopt to the commercially available dyes. Although their strategy to easily convert known HTL-substrates to sulfonate form would be useful and versatile, this achievement still seems to be a simple chemical improvement of Halo-tag system. The work would be further suitable for the readership in the specific field of chemical biology.

Reviewers' comments:

Reviewer #1 (Remarks to the Author):

The authors present a practical and streamlined solution to a well-known challenge in surface protein labelling. Given its simplicity and adaptability, this method has clear potential for widespread use in cell and molecular biology.

The authors introduced a compelling method to differentiate between the surface and intracellular pools of HTP-tagged mGluR2. Nonetheless, considering that GPCR trafficking and subcellular localisation can be highly sensitive to expression levels, this limitation requires a more thorough examination. Given the application of this technique, any additional data or discussion on how overexpression affects receptor distribution is of interest.

We thank the reviewer for their positive feedback, and acknowledge that expression levels are very important. As per request, we have added a discussion on the effects of expression levels and that more in-depth studies are warranted in the future on endogenous receptor labelling. Indeed, we are in progress to have an endogenous Grm2^{HTP/HTP} CRISPR/Cas9 knock-in made, for which our dyes would be an ideal playground to learn more about native physiology of mGluR2. Given the extensive validation, we aim to report this in a future study. The discussion about expression levels is pasted below for easy reference:

GPCRs in the central nervous system, similar as to our reported Glp1r^{SNAP/SNAP} animals to demonstrate receptor localization in the brain and super-resolution imaging¹. This is needed for a full physiological understanding, as overexpression may lead to aberrant dimer or oligomer formation of GPCRs^{2,3}, alter downstream signaling properties (e.g. ERK activation⁴, pro-survival⁵, counteracting metabolic dysfunction⁶) and localization⁷.

Furthermore, the manuscript would benefit from a more complete and transparent description of the docking methodology. The authors mention the use of the “two-point attractor method” for covalent docking in AutoDock 4, which is a valid and increasingly adopted strategy originally described by Bianco et al. (2015). However, the implementation details are not fully explained in this manuscript. It would be helpful to briefly summarise the key features of the method and how it was adapted for this particular system, especially for readers who are unfamiliar with covalent docking workflows. In addition, critical parameters, such as the choice and justification of the docking grid size, the definition of the binding site, and how covalent bond formation was enforced during the simulations, were not sufficiently reported. The manuscript should also clarify whether the reported binding affinities correspond to AutoDock’s internal scoring functions. However, these scores do not represent the true physical binding free energies, which is an important limitation that should be acknowledged. Finally, the authors might briefly comment on whether alternative scoring approaches could be applied and whether different docking decoys could be expected.

We thank the reviewer for pointing out these important points regarding the computational part of our study. In our opinion, the weaknesses that Reviewer 1 pointed out have now been eliminated. We have now compared scoring functions that are more modern and robust than the previous ones. We have also pointed to the entire ensembles, narrowed the scope of interpretation of the scores, and generated and compared the ligands in a better way, and used a modern method for covalent docking instead of relying on two-point attraction. In addition, we have applied these docking algorithms also for NBD-(S)HTL. Below, we have documented everything in detail - in our opinion, everything can now be reproduced exactly with these instructions.

Computational modelling

Methods

In order to investigate the theoretical plausibility of a sulfonated HTL (SHTL) compound as a ligand with comparable or preferably increased affinity towards the HTP receptor, computational docking was performed. To this end, the isolated ligand geometries of the dye-HTL and dye-SHTL conjugates (**Supplementary Figure S2**) were first obtained from a SMILES string, and pre-optimized using OpenBabel heuristics in combination with the MMFF94 force field (conjugated gradient algorithm, 3D conformer generation setting „best“).⁸⁻¹² Subsequently, the structures were geometry optimized using the ORCA6 quantum chemistry suite.¹³ Here, the ω B97X-D3BJ/def2-TZVPP method was used with the default integration grid settings and both TightSCF and TightOPT convergence criteria. The RIJCOSX system was employed throughout with automatic auxiliary basis set selection, and to model solvation effects, the SMD method was used for water.^{14,15} After geometry convergence, the stability of the minimum was validated by calculating the vibrational frequencies using the same method, affording exclusively positive vibrational frequencies. The receptor was obtained from the RCSB database, whereas the apo structure of the HaloTag protein was taken from PDB-5UY1.¹⁶ Using the ChimeraX Dock Prep routine, the receptor was prepared by deleting solvent, ions, and non-standard residue entities, building incomplete residues from the Dunbrack rotamer library, adding hydrogens, and adjusting charges for physiological conditions.^{17,18}

All docking experiments were performed using the GNINA 1.1 docking program, whereas the empirical scoring function „Vinardo“ was employed, in tandem with the CNN model „general_default2018“, used for re-scoring of the final poses.^{19,20} All experiments used an exhaustiveness of 128 to search for minima, whereas a total of 50 output poses was requested. To establish reproducibility, the random seed was set to 42. The search box was defined using the full receptor geometry, whereas the autobox method was used to fully enclose the receptor including 5.0 Å padding on all sides, and allowing for automatic box extension. Both ligands were docked to the receptor using the standard docking routine, affording models for the initial non-covalent association, as well as using the covalent docking approach implemented in GNINA. In the latter, the internal binding heuristics were used to simulate a covalent bond between the carboxylate of residue ASP106 and the terminal carbon of HTL and SHTL, respectively. Before covalent docking, the terminal chloride was exchanged for hydrogen, and the geometry of the ligand was re-optimized using the aforementioned DFT geometry optimization settings, but exclusively permitting hydrogen atom movement. To allow for comprehensive sampling of the conformational space, in the covalent docking, the residue ASP106 was granted torsional flexibility.

The ensemble score distributions of both ligands for both docking experiments are depicted in **Supplementary Figure S3**. It should be noted that all presented docking scores are not to be interpreted thermodynamically, but rather as descriptive parameters for the relative behaviour of both ligands. The empirical scoring function is a well-established and robust tool for screening, employing systematic treatment of different energetic terms, whereas the CNN re-scoring function is a convolutional neural network trained to estimate affinity from complex and non-linear features, implicitly encoding solvent exposure, cooperativity and complex interaction patterns that are beyond the reach of empirical scoring functions.

Selection of the best global binding poses was guided by a maximized CNN affinity score, the best poses for both ligands are depicted in **Supplementary Figure S4**. To further investigate the total ensemble of binding modes and compare both pose-score correlation and structural diversity, a principal component analysis (PCA) of all atomic cartesian coordinates for each ligand pose was performed. The first two principal components (selected by highest explained variance) are shown in **Supplementary Figure S5**.

A comprehensive study of the structural stability of the afforded complexes could be complemented by free energy perturbative methods or MM-PB/GBSA calculations on the non-covalent ensembles, as well as molecular dynamics simulations and subsequent stability evaluations for the covalently conjugated constructs (**Supplementary Figure S6**).

Statistical Analysis

To compare the ligand behaviour for both dye-HTL and dye-SHTL conjugates, several statistical tests were performed on the presented data. All tests were calculated separately for (i) the *non-covalent* and (ii) the *covalent* docking ensembles, each evaluated with the two scoring modes *Empirical* and *CNN*. For each combination of ensembles and scores, two separate questions were addressed. First, the correlation of the estimated affinity scores were tested for correlation with the principal conformational variation (PC1 + PC2) of the ligand geometries, which was assessed by multivariate least-squares regression of the affinity scores against the first two principal components. Second, the ensemble affinity scores were compared for both ligands to test for statistically significant deviations. To this end, a two-sided Mann-Whitney U test was employed, affording a non-parametric comparison of the ligand ensembles.

The results for the correlation analysis are presented in **Supplementary Table 1** and **3**, whereas the Mann-Whitney U test is shown in **Supplementary Table 2** and **4**.

The CNN re-scoring revealed strong and highly significant correlations between ligand conformational variation and estimated affinity for both ligands in the non-covalent ensemble ($R = 0.92$ for TMR-SHTL; $R = 0.46$ for TMR-HTL) and retains moderate correlation in the covalent ensembles. The empirical scoring shows weak to moderate correlations throughout.

For the non-covalent ensemble, the CNN affinity scores of TMR-SHTL are shifted significantly towards more favorable values than those of TMR-HTL ($U = 790$, $p = 0.0015$), whereas the empirical scoring revealed no significant difference ($U = 1375$, $p = 0.39$). In the covalent ensembles, both scoring functions indicated a pronounced score increase for TMR-SHTL (CNN $p < 10^{-13}$; Empirical $p < 10^{-8}$).

These results collectively demonstrate that TMR-SHTL is expected to be accommodated at least as well as TMR-HTL, and, under the more expressive CNN model, consistently exhibits superior predicted binding across both binding protocols.

For the NBD conjugates, in the non-covalent ensemble, CNN re-scoring showed a moderate and statistically significant correlation with conformational variation for NBD-SHTL ($R = 0.45$, $p = 0.001$), whereas NBD-HTL exhibited weaker but still significant correlation ($R = 0.30$, $p = 0.037$). In contrast, empirical scoring yielded only weak and non-significant trends for both ligands. In the covalent ensembles, the empirical scoring showed the strongest overall correlation ($R = 0.66$ for NBD-HTL, $R = 0.38$ for NBD-SHTL), while CNN correlations were weaker but still significant for NBD-SHTL ($R = 0.37$, $p = 0.0096$).

The Mann-Whitney U tests resolve a clear and highly significant shift toward more favorable scores for NBD-SHTL compared to NBD-HTL. In the covalent ensembles, this separation is even more pronounced for both scoring methods.

Taken together, these findings indicate that SHTL substrates are predicted to bind at least as well as HTL substrates. The CNN re-scoring model consistently identifies SHTL as the stronger binder in both non-covalent and covalent ensembles, mirroring the same trends for TMR and NBD, reinforcing HTP's general tolerance for general sulfonation of dye-HTL substrates.

A.i

A.ii

B.i

B.ii

C.i

C.ii

D.i

D.ii

Supplementary Figure S2: Topology and optimized geometry of the studied ligands. The molecular topology of TMR-HTL (A) and TMR-SHTL (B), as well as NBD-HTL (C) and NBD-SHTL (D) is depicted in (i), whereas (ii) shows the optimized geometries using ω B97X-D3BJ/def2-TZVPP[SMD(water)].

Supplementary Figure S3: Score Distributions. Empirical affinities for all poses for all ensembles of ligands, as obtained using the Vinardo scoring function. For each ensemble, all poses were re-scored using the GNINA CNN re-scoring model. (shown are all individual scores for each pose as scattered points, as well as boxplots, where the box resembles the central quartiles including the median line, and the whiskers show minimum and maximum, respectively).

Supplementary Figure S4: Best Docking Conformations. For each ensemble, the CNN scores were used to rank all obtained poses and select the best geometry as the one having the highest affinity score. Each depiction shows the receptor (PDB-5UY1, HaloTag) as semi-transparent cartoon, as well as the ligand conformation (cyan). For each model, the residues within a 4.5 Å radius around the ligand are shown as wireframes. In the covalent poses, the conjugated residue ASP106 is rendered in cyan as well.

Supplementary Figure S5: Principal components for all ensembles. For each ensemble, all obtained poses were subjected to principal component analysis, whereas the cartesian coordinates of each ligand geometry were used as descriptors. Each pose is depicted as a point in the plane of the first two principal components, as selected by their respective explained variance (given on each axis in percent).

Supplementary Figure S6: Conformer Ensembles after Docking. For each ligand, the non-covalently and covalently docked conformer ensembles are depicted. The receptor (PDB-5UY1, HaloTag) is rendered as semi-transparent cartoon, whereas the ligand (including the conjugated ASP106) is rendered as stick models in cyan.

Supplementary Table 1. Multivariate regression results for correlation analysis of affinity scores vs. principal conformational variations, as obtained for all modelled ensembles of TMR-HTL and TMR-SHTL.

Docking Mode	Scoring	Ligand	R (affinity vs. PC1+PC2)	p -value
Non-Covalent	Empirical	TMR-HTL	0.282	4.72×10^{-2}
		TMR-SHTL	0.428	1.94×10^{-3}
	CNN	TMR-HTL	0.458	8.24×10^{-4}
		TMR-SHTL	0.918	6.93×10^{-21}
Covalent	Empirical	TMR-HTL	0.090	0.546
		TMR-SHTL	0.195	0.176
	CNN	TMR-HTL	0.288	4.99×10^{-2}
		TMR-SHTL	0.291	4.01×10^{-2}

Supplementary Table 2. Mann-Whitney U test results for paired comparisons between both ligands for all modelled ensembles of TMR-HTL and TMR-SHTL.

Docking Mode	Scoring	Mann-Whitney U	p -value
Non-Covalent	Empirical	1375	3.91×10^{-1}
	CNN	790	1.54×10^{-3}
Covalent	Empirical	1981	6.08×10^{-9}
	CNN	114	1.93×10^{-14}

Supplementary Table 3. Multivariate regression results for correlation analysis of affinity scores vs. principal conformational variations, as obtained for all modelled ensembles of NBD-HTL and NBD-SHTL.

Docking Mode	Scoring	Ligand	R (affinity vs. PC1+PC2)	p -value
Non-Covalent	Empirical	NBD-HTL	0.249	8.45×10^{-2}
		NBD-SHTL	0.185	1.98×10^{-1}
	CNN	NBD-HTL	0.299	3.71×10^{-2}
		NBD-SHTL	0.450	1.03×10^{-3}
Covalent	Empirical	NBD-HTL	0.656	7.16×10^{-3}
		NBD-SHTL	0.379	3.19×10^{-7}
	CNN	NBD-HTL	0.208	1.52×10^{-1}
		NBD-SHTL	0.366	9.60×10^{-3}

Supplementary Table 4. Mann-Whitney U test results for paired comparisons between both ligands for all modelled ensembles of NBD-HTL and NBD-SHTL.

Docking Mode	Scoring	Mann-Whitney U	p -value
Non-Covalent	Empirical	1867	7.14×10^{-6}
	CNN	455	7.23×10^{-8}
Covalent	Empirical	2179	3.68×10^{-12}
	CNN	80	1.75×10^{-15}

Reviewer #2 (Remarks to the Author):

In this manuscript, the authors report a method they describe as “a general one-step protocol to generate impermeable fluorescent HaloTag substrates,” with demonstrations in super-resolution imaging. While the biological applications are well-executed and add value, the chemical aspect of the work does not present a significant advance over existing methods. The approach primarily involves incorporating a charged sulfonic acid group into the fluorophore-conjugated HaloTag ligand—a strategy that has already been explored in various forms, including placing the HaloTag ligand and the anionic group at different positions on the fluorophore. The modification of rhodamine dyes with functional groups such as sulfonates is a well-established technique and can often be done without substantially altering the photophysical properties of the dyes. As such, the chemical novelty of this method appears limited, and its distinction from previously published approaches is unclear.

We thank the reviewer to value the biological part, and respectfully but strongly disagree with the notion that our approach lacks novelty. Indeed, sulfonation is a standard strategy which has been applied directly to dyes, which requires many synthetic steps. In stark contrast, our strategy of sulfonating the amine (new Figure 3) and amide bond of HTLs has not been done before (thereby reflecting novelty), avoids direct modification to the dye itself, and the beauty lies in the fact that commercially available substrates may be used, making this technique available to researchers without chemical infrastructure (hoods, HPLC, etc.). We have added NBD to the portfolio of sulfonation and used it for docking, as well as for cellular confocal imaging, demonstrating generalizability further. The text and Figure is pasted below:

Rhodamines linked by peptide bonds to SLP ligands remain often the dye and connection type of choice for HTP labelling and cellular imaging²¹, however, we wondered if our protocol is more general and amenable to completely scaffolds. Nitrobenzodiazaisooxazole (NBD) are classically fluorophores in the green spectrum with a vastly different molecular structure.²² They have recently been upgraded to span a larger color spectrum and become known as the SCOTfluors (ref²³), and furthermore have been used on solid phase support for peptide synthesis^{24,25}, so we picked this dye class for testing our protocol to also expand our strategy towards amine derivatization (**Fig. 3A**). Before commencing with the synthesis, we docked NBD-HTL (**Fig. 3B**) and NBD-SHTL (**Fig. 3C**) covalently to the HaloTag by the methods as described before, and observed that both structures are sterically tolerated *in silico*. NBD-HTL was then straightforwardly synthesized by using NBD-Cl and HTL-NH₂ in EtOH to precipitate analytically clean permeable ligand, which is subjected to treatment with NaH in DMF, and sulfonation by propylene sulfone to yield NBD-SHTL (**Fig. 3D**). Since the amine is defining the photophysical properties of NBD, we recorded an expected red-shift in its excitation when bis-alkylated, however, emission remains comparable ($\lambda_{Exc/Em}$ (NBD-HTL) = 479 / 552 nm; $\lambda_{Exc/Em}$ (NBD-SHTL) = 498 / 546 nm) (**Fig. 3E**). We confirmed that NBD-(S)HTL does label recombinant HTP (**Fig. 3F**) by full protein mass spectrometry, and then incubated both ligands with HEK293T cells for confocal imaging. While NBD-HTL is uptaken into cells unspecifically, we did not observe any fluorescence signal when using NBD-SHTL (**Fig. S8**). Next, we transfected HEK293T cells with the aforementioned, SNAP-TM-HTP and HTP-TM-SNAP systems, and indeed, observed surface labelling of NBD-SHTL when the HTP is extracellularly exposed (**Fig. 3G**) and intracellular labelling for NBD-HTL (**Fig. 3H**). BG-JF₆₄₆ was again used for SNAP tagging.

Figure 3: Sulfonation on an amine linked NBD dye. **A)** Expansion of the approach from red/far-red amide conjugated HTL dyes to green and amine linked NBD. **B, C)** Modelling of the HaloTag Protein (HTP) bound to NBD-(S)HTL. **D)** Synthesis of NBD-HTL and NBD-SHTL. **E)** Excitation and emission profiles of NBD-HTL and NBD-SHTL. $n=3$. **F)** *In vitro* protein labelling of apo-HTP confirms binding by full protein mass spectrometry. **G)** Confocal images of HEK293T expressing HTP-TM-SNAP. SNAP labelled with BG-JF₆₄₆; NBD-HTL stains cells unspecifically while NBD-SHTL labels extracellular HTP. **H)** As for (G) but transfection with SNAP-TM-HTP. Imaging was performed for all conditions in $N=3$ with $n=5$.

Supplementary Figure S8: Control experiments for confocal imaging using NBD-(S)HTL in non-transfected HEK293T cells.

This main point also highlights that the goal of the approach is to avoid altering photophysical properties. We do agree that the distinction to other methods has come short, and as such, add a new Figure S1 in which we challenge our hypothesis that sulfonation if the alkane linker abolishes HTP binding, and in addition provide a broader explanation and comparison, citing the recent developments in the field, as below:

RESULTS PART

A more general and effective strategy would offer a universal solution for all dyes. For instance, we recently described a modified version of the SNAP ligand in which the benzylguanane (BG) leaving group carries a

negative charge due to incorporation of a sulfonate on the 8-position of guanine, resulting in 'SBG substrates' that are released upon covalent reaction (**Fig. 1A**, right and **Fig. 1B**).²⁶ Unfortunately, this approach is not possible for the HaloTag system, since the HTL leaving group is a chloride anion, and chemical oxidation would result in hypochlorite (RCIO), chlorite (RCIO₂), chlorate (RCIO₃) or perchlorate (RCIO₄), which all would maintain an overall neutral charge. Nevertheless, we challenged ourselves and synthesized negatively charged TMR-HTL-OSO₃⁻ and TMR-HTL-OPO₃²⁻ that bear an alkyl sulfonate and phosphonate, respectively, instead of the alkyl chloride as a leaving group (**Fig. S1A**). While the phosphonate is not stable within minutes in PBS as assessed by LCMS, the sulfonate did not show decomposition over night (**Fig. S1B**). Still, and as expected, TMR-HTL-sulfonate did not covalently label recombinant HTP as assessed by full protein mass spectrometry (**Fig. S1C**).

Supplementary Figure S1: A sulfonated and phosphonated HTL does not label the HTP. (A) Synthesis of **TMR-HTL^{OH}**, **TMR-HTL^{SO4}** and **TMR-HTL^{PO4}** by displacement of the chloride atom with a hydroxy group using silver carbonate, in situ deprotection of the Boc group and amide bond formation using **TMR-NHS**. The terminal charges are installed by using either sulfonylchloride or phosphooychloride, followed by quenching the reaction with water. **B)** **TMR-HTL^{SO4}**, but not **TMR-HTL^{PO4}**, is stable in PBS over hours as assessed by LCMS analysis. **C)** Recombinant HTP incubated **TMR-HTL^{SO4}** is not covalently labelled, as assessed by full protein mass spectrometry.

DISCUSSION PART

The development of bright (and impermeable) fluorophores for microscopy has garnered increasing interest,^{27–29} particularly in the area of shadow imaging^{30,31} and protein labelling^{1,32–34}. In this study, we aimed to render existing and commercially available dye-HTL substrates impermeable to a cell's plasma membrane for a targeted approach, similar to strategies that have been applied to coumarin-arachidonic acid conjugates for uncaging a

signalling lipid³⁵, BODIPY impermeabilization to tune cellular compartmentalization³⁶, Heidelberg Dyes for turn-on tetrazine click chemistry³⁷ and creating an impermeable version for the fluorescence-activating and absorption-shifting tag (FAST)³⁸. In our case, we applied a quick and straightforward one step protocol on available rhodamine scaffolds, since these dye-HTL molecules only bear one acidic proton, and furthermore, in basic solutions form a non-fluorescent spirolactone form, preventing off-target alkylation on the carboxylate. This is unprecedented for HaloTagging and distinguishes itself from other approaches, most importantly from equipping the dye with a charged group, which needs *de novo* synthesis, adding additional synthetic steps prior to fusing the dye with an SLP ligand (e.g. for rhodamine 6G (5 steps)³⁹; for JF₆₃₅ⁱ and Sulfo549/646 (3 and 5 steps, respectively)^{32,40}). It is also in stark contrast to using commercially available impermeable dyes (e.g. Alexa Fluor 594, ATTO 532, SulfoCy3/5) and linking them to HTLs, since this would require coupling and HPLC purification, which we circumvent with an *in situ* protocol. We obtained *cis*- and *trans*-amide mixtures (as per ¹H NMR, see SI), and observed full protein labelling by mass spectrometry using recombinant HTP, without any significant difference in labelling kinetics *via* fluorescence polarization compared to non-sulfonated HTL substrates. The majority of dye-HTL reagents should be amenable to this method, however, the approach would not be applicable to certain scaffolds with other reactive attachment sites, including biotin, dyes with nucleophilic sites, for instance NH-anilines (e.g. ATTO 532, rhodamine 6G, SiR595) or hydroxy groups (e.g. Oregon Green) as it would abolish its pharmacology (e.g. biotin will not bind streptavidin) or change the fluorescent properties (e.g. red-shift for ATTO 532, rhodamine 6G, SiR595; extinguished fluorescence for *O*-alkylated Oregon Green). With such dyes not being attractive for our method, we chose nitrobenzodiazaisooxazole (NBD) was chosen to expand the repertoire, and provide further generalizability of sulfonation, adding a dye to the red/far-red color palette by showing fluorescence in the green, and secondly, broadening the protocol to other chemical moieties by addressing an amine (and not an amide) as the sulfonation site.

The authors conducted extensive biological experiments to validate their approach; however, the applications demonstrated are fairly typical for super-resolution fluorescence microscopy.

We thank the reviewer for acknowledging the extensive biology, and we used transduced primary neurons to image synapses and quantify receptor localization to nanodomains, which we think is beyond being typical for chemical biology studies, which usually halts at using immortalized cell lines. In fact, our demonstration led to a surprising finding that surface and intracellular mGluR2 pools can have distinct localization relative to the synapse. To underpin this, we have added to the discussion as follows, highlighting the use of STED nanoscopy for novel biology up until today:

To this end, this represents state-of-the-art neuronal imaging, as STED has recently helped to understand nanoarchitecture of synapses in general^{41,42}, and in particular on disease-relevant proteins, such as Munc13-1 (involved in the exocytotic machinery)⁴³, on SYP1/SYT1/VGLUT (markers for synaptic vesicles)⁴⁴, Tenm3-Lphn3 complexes (that reconstitute synaptic junctions)⁴⁵, Blobby (an active zone assembly protein necessary for memory formation in *D. melanogaster*)⁴⁶ and NMDA receptors (in a trans-synaptic context)⁴⁷.

In addition, we have further highlighted that we used two-color STED, which further demonstrates that we did not halt at 'standard procedure', and have emphasized this by updating Figure 5 (new Figure 5D) with a new subpanel that shows higher resolution of Bassoon using STED on super-resolution imaging of processes. The updated full Figure is pasted below:

Figure 5: STED super-resolution imaging of surface HTP(:S-SiR-d12)-mGluR2. **A)** Confocal and STED images of HTP(:S-SiR-d12)-mGluR2 transduced neurons. **B)** Line scan profile of a process comparing confocal to STED performance, yielding a resolution of 134 nm across the ultrastructure. **C)** Three color confocal and dual color STED with zoom in of the process reported in (B). **D)** Line scan profile of the pre-synaptic marker Bassoon comparing confocal to STED performance allows the reporting of immunostained proteins additional to SHTL labelling.

Additionally, the deuterated rhodamine dyes (TMR-d12 and SiR-d12) show only marginal improvement over their non-deuterated counterparts and do not offer a substantial performance benefit for super-resolution imaging.

The point of the study is not to demonstrate the superiority of deuterated TMR and SiR – this we have published some years ago (Roßmann et al., Chem. Sci. 2022). Indeed, we aim to show that our general sulfonation approach allows us to use the best-in-class dyes and to simply render them impermeable. Our argument has never been that dye performance would improve from our approach, rather that dye performance would be unaltered, a critical difference that speaks to the orthogonality of our chemical modification strategy.

Therefore, I rejected this work for publication in Nature Communications.

We are sorry to have read this, and hope to convince the reviewer that 1) our method has added novelty, 2) is expandable to a broad repertoire of HTL linked dyes with different chemistry, and 3) that we have worked hard to show the potential of our approach in state-of-the-art super-resolution microscopy on a receptor crucially involved in pathologies, such as anxiety and depression.

Reviewer #3 (Remarks to the Author):

In this manuscript, the authors described a new strategy to make HaloTag ligand (HTL) impermeable to cell membrane via simple chemical synthetic method, so that HaloTag-conjugated proteins (HTP) can be labeled only on the cellular surface. They demonstrated selective surface protein labeling to live HEK293 cells and hippocampal neurons genetically modified with HTP. The authors also proposed a simple and efficient protocol for synthesizing cell-impermeable HTL from known HTLs. The experimental procedures seem technically sound and the results look appropriately processed and presented. The authors demonstrated synaptic mGluR2 localization and found the relatively diverse distribution of the receptors in detail, by means of their new sulfonated HTP (SHTP) and super resolution fluorescence microscopy. They also showed their strategy is applicable and useful to conventional confocal microscopy. Their approach for clear and selective labeling to membrane surface proteins would be successful and useful to track the proteins-of-interest, and the combination of a simple chemical modification on the HTL with genetic manipulation of the target cells/tissue is attractive in a view point of chemical biology. However, the basic technologies employed in this study would be well established except their molecular design and modification reaction to introduce sulfonate groups with sulfone. Although the optimization of the position of medication and the reaction conditions might be not straightforward, it would be rather chemical issue how the utility of sulfonate groups increases with simple introduction procedures for many kinds of useful dyes. The method the authors proposed has still limitation to synthesize sulfonated ligands such as stability for strong base reagents, while there are already several strategies to develop sulfonated fluorescence dye parts as the authors mentioned, even though they also have their own limitations. The work is interesting and attractive, but rather suitable for the specific community of organic chemistry and chemical biology.

We thank the reviewer for their positive assessment, yet it is not entirely clear what the critique points are. We read into the review that 1) the generalizability and 2) the stability of the dyes are a concern, thereby raising similar points as reviewer #2. Addressing the latter:

We tested the claim that stability of other dyes may result in a problem and tested our approach on NBD-HTL, in which the dye survives the conditions at room temperature over night (55% isolated yield after HPLC purification). We would like to point out, however, that the main point is that commercial substrates may be used.

Addressing the first point, we have synthesized and used NBD-SHTL successfully versus NBD-HTL in confocal imaging (new Figure 3), and have added to the results and discussion for reviewer #2, pasted below for easy reference:

RESULTS PART

A more general and effective strategy would offer a universal solution for all dyes. For instance, we recently described a modified version of the SNAP ligand in which the benzylguanine (BG) leaving group carries a negative charge due to incorporation of a sulfonate on the 8-position of guanine, resulting in 'SBG substrates' that are released upon covalent reaction (**Fig. 1A**, right and **Fig. 1B**).²⁶ Unfortunately, this approach is not possible for the HaloTag system, since the HTL leaving group is a chloride anion, and chemical oxidation would result in hypochlorite (RCIO), chlorite (RCIO₂), chlorate (RCIO₃) or perchlorate (RCIO₄), which all would maintain an overall neutral charge. Nevertheless, we challenged ourselves and synthesized negatively charged TMR-HTL-OSO₃⁻ and TMR-HTL-OPO₃²⁻ that bear an alkyl sulfonate and phosphonate, respectively, instead of the alkyl chloride as a leaving group (**Fig. S1A**). While the phosphonate is not stable within minutes in PBS as assessed by LCMS, the sulfonate did not show decomposition over night (**Fig. S1B**). Still, and as expected, TMR-HTL-sulfonate did not covalently label recombinant HTP as assessed by full protein mass spectrometry (**Fig. S1C**).

Supplementary Figure S1: A sulfonated and phosphonated HTL does not label the HTP. (A) Synthesis of **TMR-HTL^{OH}**, **TMR-HTL^{SO4}** and **TMR-HTL^{PO4}** by displacement of the chloride atom with a hydroxy group using silver carbonate, in situ deprotection of the Boc group and amide bond formation using **TMR-NHS**. The terminal charges are installed by using either sulfonylchloride or phosphooychloride, followed by quenching the reaction with water. **B)** **TMR-HTL^{SO4}**, but not **TMR-HTL^{PO4}**, is stable in PBS over hours as assessed by LCMS analysis. **C)** Recombinant HTP incubated **TMR-HTL^{SO4}** is not covalently labelled, as assessed by full protein mass spectrometry.

Rhodamines linked by peptide bonds to SLP ligands remain often the dye and connection type of choice for HTP labelling and cellular imaging²¹, however, we wondered if our protocol is more general and amenable to completely scaffolds. Nitrobenzodiazaisooxazole (NBD) are classically fluorophores in the green spectrum with a vastly different molecular structure.²² They have recently been upgraded to span a larger color spectrum and become known as the SCOTfluors (ref²³), and furthermore have been used on solid phase support for peptide synthesis^{24,25}, so we picked this dye class for testing our protocol to also expand our strategy towards amine derivatization (**Fig. 3A**). Before commencing with the synthesis, we docked NBD-HTL (**Fig. 3B**) and NBD-SHTL (**Fig. 3C**) covalently to the HaloTag by the methods as described before, and observed that both structures are sterically tolerated *in silico*. NBD-HTL was then straightforwardly synthesized by using NBD-Cl and HTL-NH₂ in EtOH to precipitate analytically clean permeable ligand, which is subjected to treatment with NaH in DMF, and sulfonation by propylene sulfone to yield NBD-SHTL (**Fig. 3D**). Since the amine is defining the photophysical properties of NBD, we recorded an expected red-shift in its excitation when bis-alkylated, however, emission remains comparable (λ_{Exc}/Em (NBD-HTL) = 479 / 552 nm; λ_{Exc}/Em (NBD-SHTL) = 498 / 546 nm) (**Fig. 3E**). We confirmed that NBD-(S)HTL does label recombinant HTP (**Fig. 3F**) by full protein mass spectrometry, and then incubated both ligands with HEK293T cells for confocal imaging. While NBD-HTL is uptaken into cells

unspecifically, we did not observe any fluorescence signal when using NBD-SHTL (**Fig. S8**). Next, we transfected HEK293T cells with the aforementioned, SNAP-TM-HTP and HTP-TM-SNAP systems, and indeed, observed surface labelling of NBD-SHTL when the HTP is extracellularly exposed (**Fig. 3G**) and intracellular labelling for NBD-HTL (**Fig. 3H**). BG-JF₆₄₆ was again used for SNAP tagging.

Figure 3: Sulfonation on an amine linked NBD dye. **A)** Expansion of the approach from red/far-red amide conjugated HTL dyes to green and amine linked NBD. **B, C)** Modelling of the HaloTag Protein (HTP) bound to NBD-(S)HTL. **D)** Synthesis of NBD-HTL and NBD-SHTL. **E)** Excitation and emission profiles of NBD-HTL and NBD-SHTL. $n=3$. **F)** *In vitro* protein labelling of apo-HTP confirms binding by full protein mass spectrometry. **G)** Confocal images of HEK293T expressing HTP-TM-SNAP. SNAP labelled with BG-JF₆₄₆; NBD-HTL stains cells unspecifically while NBD-SHTL labels extracellular HTP. **H)** As for (G) but transfection with SNAP-TM-HTP. Imaging was performed for all conditions in $N=3$ with $n=5$.

Supplementary Figure S8: Control experiments for confocal imaging using NBD-(S)HTL in non-transfected HEK293T cells.

DISCUSSION PART

The development of bright (and impermeable) fluorophores for microscopy has garnered increasing interest,^{27–29} particularly in the area of shadow imaging^{30,31} and protein labelling^{1,32–34}. In this study, we aimed to render existing and commercially available dye-HTL substrates impermeable to a cell's plasma membrane for a targeted approach, similar to strategies that have been applied to coumarin-arachidonic acid conjugates for uncaging a signalling lipid³⁵, BODIPY impermeabilization to tune cellular compartmentalization³⁶, Heidelberg Dyes for turn-on tetrazine click chemistry³⁷ and creating an impermeable version for the fluorescence-activating and absorption-shifting tag (FAST)³⁸. In our case, we applied a quick and straightforward one step protocol on available rhodamine scaffolds, since these dye-HTL molecules only bear one acidic proton, and furthermore, in basic solutions form a non-fluorescent spirolactone form, preventing off-target alkylation on the carboxylate. This is unprecedented for HaloTagging and distinguishes itself from other approaches, most importantly from equipping the dye with a charged group, which needs *de novo* synthesis, adding additional synthetic steps prior to fusing the dye with an SLP ligand (e.g. for rhodamine 6G (5 steps)³⁹; for JF₆₃₅i and Sulfo549/646 (3 and 5 steps, respectively)^{32,40}). It is also in stark contrast to using commercially available impermeable dyes (e.g. Alexa Fluor 594, ATTO 532, SulfoCy3/5) and linking them to HTLs, since this would require coupling and HPLC purification, which we circumvent with an *in situ* protocol. We obtained *cis*- and *trans*-amide mixtures (as per ¹H NMR, see SI), and observed full protein labelling by mass spectrometry using recombinant HTP, without any significant difference in labelling kinetics *via* fluorescence polarization compared to non-sulfonated HTL substrates. The majority of dye-HTL reagents should be amenable to this method, however, the approach would not be applicable to certain scaffolds with other reactive attachment sites, including biotin, dyes with nucleophilic sites, for instance NH-anilines (e.g. ATTO 532, rhodamine 6G, SiR595) or hydroxy groups (e.g. Oregon Green) as it would abolish its pharmacology (e.g. biotin will not bind streptavidin) or change the fluorescent properties (e.g. red-shift for ATTO 532, rhodamine 6G, SiR595; extinguished fluorescence for *O*-alkylated Oregon Green). With such dyes not being attractive for our method, we chose nitrobenzodiazaisooxazole (NBD) was chosen to expand the repertoire, and provide further generalizability of sulfonation, adding a dye to the red/far-red color palette by showing fluorescence in the green, and secondly, broadening the protocol to other chemical moieties by addressing an amine (and not an amide) as the sulfonation site.

REFERENCES

1. Ast, J. *et al.* Revealing the tissue-level complexity of endogenous glucagon-like peptide-1 receptor expression and signaling. *Nat Commun* **14**, 301 (2023).
2. Sleno, R. & Hébert, T. E. The Dynamics of GPCR Oligomerization and Their Functional Consequences. in *International Review of Cell and Molecular Biology* vol. 338 141–171 (Elsevier, 2018).
3. Ferré, G. *et al.* Structure and dynamics of G protein-coupled receptor-bound ghrelin reveal the critical role of the octanoyl chain. *Proc Natl Acad Sci USA* **116**, 17525–17530 (2019).
4. Lee, H. *et al.* The effect of mGluR2 activation on signal transduction pathways and neuronal cell survival. *Brain Res* **1249**, 244–250 (2009).
5. Olaniru, O. E. *et al.* SNAP-tag-enabled super-resolution imaging reveals constitutive and agonist-dependent trafficking of GPR56 in pancreatic β -cells. *Molecular Metabolism* **53**, 101285 (2021).
6. Sveidahl Johansen, O. *et al.* Lipolysis drives expression of the constitutively active receptor GPR3 to induce adipose thermogenesis. *Cell* **184**, 3502-3518.e33 (2021).
7. Blythe, E. E. & Von Zastrow, M. β -Arrestin-independent endosomal cAMP signaling by a polypeptide hormone GPCR. *Nat Chem Biol* **20**, 323–332 (2024).
8. Weininger, D. SMILES, a chemical language and information system. 1. Introduction to methodology and encoding rules. *J. Chem. Inf. Comput. Sci.* **28**, 31–36 (1988).
9. Weininger, D., Weininger, A. & Weininger, J. L. SMILES. 2. Algorithm for generation of unique SMILES notation. *J. Chem. Inf. Comput. Sci.* **29**, 97–101 (1989).
10. Weininger, D. SMILES. 3. DEPICT. Graphical depiction of chemical structures. *J. Chem. Inf. Comput. Sci.* **30**, 237–243 (1990).
11. O’Boyle, N. M. *et al.* Open Babel: An open chemical toolbox. *Journal of Cheminformatics* **3**, 33 (2011).
12. Halgren, T. A. Merck molecular force field. I. Basis, form, scope, parameterization, and performance of MMFF94. *Journal of Computational Chemistry* **17**, 490–519 (1996).
13. Neese, F. Software Update: The ORCA Program System—Version 6.0. *WIREs Computational Molecular Science* **15**, e70019 (2025).
14. Neese, F. An improvement of the resolution of the identity approximation for the formation of the Coulomb matrix. *J Comput Chem* **24**, 1740–1747 (2003).
15. Neese, F., Wennmohs, F., Hansen, A. & Becker, U. Efficient, approximate and parallel Hartree–Fock and hybrid DFT calculations. A ‘chain-of-spheres’ algorithm for the Hartree–Fock exchange. *Chemical Physics* **356**, 98–109 (2009).
16. Liu, Y. *et al.* The Cation– π Interaction Enables a Halo-Tag Fluorogenic Probe for Fast No-Wash Live Cell Imaging and Gel-Free Protein Quantification. *Biochemistry* **56**, 1585–1595 (2017).
17. Meng, E. C. *et al.* UCSF CHIMERAX: Tools for structure building and analysis. *Protein Science* **32**, e4792 (2023).
18. Shapovalov, M. V. & Dunbrack, R. L. A smoothed backbone-dependent rotamer library for proteins derived from adaptive kernel density estimates and regressions. *Structure* **19**, 844–858 (2011).
19. McNutt, A. T. *et al.* GNINA 1.0: molecular docking with deep learning. *J Cheminform* **13**, 43 (2021).
20. Sunseri, J. & Koes, D. R. Virtual Screening with Gnina 1.0. *Molecules* **26**, 7369 (2021).
21. Cook, A., Walterspiel, F. & Deo, C. HaloTag-Based Reporters for Fluorescence Imaging and Biosensing. *ChemBioChem* **24**, e202300022 (2023).
22. Jiang, C. *et al.* NBD-based synthetic probes for sensing small molecules and proteins: design, sensing mechanisms and biological applications. *Chem Soc Rev* **50**, 7436–7495 (2021).
23. Benson, S. *et al.* SCOTfluors: Small, Conjugatable, Orthogonal, and Tunable Fluorophores for In Vivo Imaging of Cell Metabolism. *Angew Chem Int Ed* **58**, 6911–6915 (2019).
24. De Moliner, F. *et al.* Small Fluorogenic Amino Acids for Peptide-Guided Background-Free Imaging. *Angew Chem Int Ed* **62**, e202216231 (2023).
25. Oyama, T. *et al.* Late-stage peptide labeling with near-infrared fluorogenic nitrobenzodiazoles by manganese-catalyzed C–H activation. *Chem. Sci.* **14**, 5728–5733 (2023).
26. Poc, P. *et al.* Interrogating surface versus intracellular transmembrane receptor populations using cell-impermeable SNAP-tag substrates. *Chem. Sci.* **11**, 7871–7883 (2020).
27. Schnermann, M. J. & Lavis, L. D. Rejuvenating old fluorophores with new chemistry. *Current Opinion in Chemical Biology* **75**, 102335 (2023).
28. Jiang, G. *et al.* Chemical Approaches to Optimize the Properties of Organic Fluorophores for Imaging and Sensing. *Angew Chem Int Ed* **63**, e202315217 (2024).
29. Munan, S., Chang, Y.-T. & Samanta, A. Chronological development of functional fluorophores for bio-imaging. *Chem. Commun.* **60**, 501–521 (2024).
30. Tønnesen, J., Inavalli, V. V. G. K. & Nägerl, U. V. Super-Resolution Imaging of the Extracellular Space in Living Brain Tissue. *Cell* **172**, 1108-1121.e15 (2018).
31. Velicky, P. *et al.* Dense 4D nanoscale reconstruction of living brain tissue. *Nat Methods* **20**, 1256–1265 (2023).
32. Birke, R. *et al.* Sulfonated red and far-red rhodamines to visualize SNAP- and Halo-tagged cell surface proteins. *Org Biomol Chem* **20**, 5967–5980 (2022).

33. Lee, J. *et al.* Distinct beta-arrestin coupling and intracellular trafficking of metabotropic glutamate receptor homo- and heterodimers. *Sci. Adv.* **9**, eadi8076 (2023).
34. Calebiro, D. *et al.* Single-molecule analysis of fluorescently labeled G-protein-coupled receptors reveals complexes with distinct dynamics and organization. *Proc. Natl. Acad. Sci. U.S.A.* **110**, 743–748 (2013).
35. Nadler, A. *et al.* Exclusive photorelease of signalling lipids at the plasma membrane. *Nat Commun* **6**, 10056 (2015).
36. Kand, D. *et al.* Water-Soluble BODIPY Photocages with Tunable Cellular Localization. *J. Am. Chem. Soc.* **142**, 4970–4974 (2020).
37. Werther, P. *et al.* Bio-orthogonal Red and Far-Red Fluorogenic Probes for Wash-Free Live-Cell and Super-resolution Microscopy. *ACS Cent. Sci.* **7**, 1561–1571 (2021).
38. Li, C. *et al.* Fluorogenic Probing of Membrane Protein Trafficking. *Bioconjugate Chem.* **29**, 1823–1828 (2018).
39. Boyarskiy, V. P. *et al.* Photostable, Amino Reactive and Water-Soluble Fluorescent Labels Based on Sulfonated Rhodamine with a Rigidized Xanthene Fragment. *Chemistry A European J* **14**, 1784–1792 (2008).
40. Jonker, C. T. H. *et al.* Accurate measurement of fast endocytic recycling kinetics in real time. *J Cell Sci* (2019) doi:10.1242/jcs.231225.
41. Arizono, M., Idziak, A. & Nägerl, U. V. Live STED imaging of functional neuroanatomy. *Nat Protoc* **20**, 2261–2285 (2025).
42. Maglione, M. & Sigrist, S. J. Imaging Synapse Ultrastructure and Organization with STED Microscopy. *Methods Mol Biol* **2910**, 135–144 (2025).
43. Kowald, M. *et al.* Endogenous SNAP-Tagging of Munc13-1 for Monitoring Synapse Nanoarchitecture. *JACS Au* **5**, 2475–2490 (2025).
44. Saal, K. A. *et al.* Heat denaturation enables multicolor X10-STED microscopy. *Sci Rep* **13**, 5366 (2023).
45. Zhang, X., Chen, X., Matúš, D. & Südhof, T. C. Reconstitution of synaptic junctions orchestrated by teneurin-latrophilin complexes. *Science* **387**, 322–329 (2025).
46. Lützkendorf, J. *et al.* Blobby is a synaptic active zone assembly protein required for memory in *Drosophila*. *Nat Commun* **16**, 271 (2025).
47. Anderson, M. C. *et al.* Trans-synaptic molecular context of NMDA receptor nanodomains. Preprint at <https://doi.org/10.1101/2023.12.22.573055> (2025).

We thank the expert reviewers for their time and assessment of our manuscript. Please find a point-to-point response below.

REVIEWERS' COMMENTS

Reviewer #1 (Remarks to the Author):

The authors have addressed all the methodological concerns raised in the previous revision round. The revised Supplementary Information now contains sufficient detail to ensure the full reproducibility of the computational workflow, including ligand optimization, docking configurations, re-scoring, and statistical analysis. The figures and tables have been significantly improved and now convincingly support the conclusions. The revision significantly strengthened the manuscript, and the claims are now well-supported.

We thank the reviewer for their support.

Reviewer #2 (Remarks to the Author):

In the revised version of the manuscript, the authors have addressed most of the comments raised by the reviewers. The authors synthesized a series of modified dyes and conducted extensive biological experiments to demonstrate a one-step protocol for generating impermeable fluorescent HaloTag substrates. The novelty of the work primarily lies in modifying the molecular structure by introducing a sulfonate anionic group into the HTL moiety rather than incorporating the anionic group directly onto the dye scaffold, which indeed simplifies the synthetic process to some extent. The study is new and of interest.

We thank the reviewer for their assessment and for pointing out the novelty of our approach.

However, this improvement does not constitute a substantial advance in the development of improved fluorescent dyes for in situ live-cell applications.

We agree, since we opted to retain the fluorescent characteristics of the dyes, and the improvement lies in the application to distinguish intra/extracellular protein pools.

Since HTL is covalently conjugated to the dye, it can essentially be regarded as part of the dye molecule. Thus, the distinction between introducing a sulfonate group onto the HTL moiety versus directly onto the dye is not particularly significant.

The significance stems for being able to assess receptor populations, and the introduction of a sulfonate group on a dye-ligand system rather than to a dye to then synthesize a dye-ligand pair makes the approach straightforward. We do also acknowledge the limitations of our study by stating in the manuscript "The majority of dye-HTL reagents should be amenable to this method, however, the approach would not be applicable to certain scaffolds with other reactive attachment sites, including biotin, dyes with nucleophilic sites, for instance NH-anilines (e.g. ATTO 532, rhodamine 6G, SiR595) or hydroxy groups (e.g. Oregon Green) as it would abolish its pharmacology (e.g. biotin will not bind streptavidin) or change the fluorescent properties (e.g. red-shift for ATTO 532, rhodamine 6G, SiR595; extinguished fluorescence for *O*-alkylated Oregon Green).".

Additionally, from an organic synthesis standpoint, incorporating a sulfonate group is not technically challenging.

The aim to introduce a sulfonate group without the need to purify the reaction mixture should indeed be not technically challenging. This is why we delineate a protocol that can be used by researchers without chemical infrastructure.

Although the authors performed numerous imaging experiments, including confocal and super-resolution microscopy, these data alone do not substantively enhance the novelty of the work.

With our many imaging experiments we showcase the robustness and versatility of the SHTL approach. Some novel aspects in the biological experiments are represented by 1) discerning mGluR2 localization in synapses, 2) super-resolution assessment of mGluR2 in processes in high definition to resolve the outer membrane of these thin filaments, 3) taming ATTO 647N's stickiness to be used at very low (1 nM) concentration in sparsely transfected cells by only 10 minute labelling, and 4) a general protocol that may be implemented by any researcher without chemical infrastructure.

Therefore, I agree with Referee 3 that this manuscript may be more suitable for a specialized journal in organic chemistry and chemical biology rather than a broader readership.

Reviewer #3 (Remarks to the Author):

In this revised manuscript, the authors politely responded to the comment from this reviewer although the comment from this reviewer contains ambiguous points. Reviewing the additional experimental results such as new Figure 3, this reviewer understands that the strategy of their work is applicable to another dye, NBD-HTL, via same synthetic methodology, and also that this strategy is easy to adopt to the commercially available dyes. Although their strategy to easily convert known HTL-substrates to sulfonate form would be useful and versatile, this achievement still seems to be a simple chemical improvement of Halo-tag system.

We thank the reviewer for acknowledging our efforts to improve our manuscript, and are happy to hear that the strategy is deemed useful and versatile. And indeed, we deliberately opted for a simple and straightforward chemical derivatization of dye-HTLs to be used in several imaging experiments.

The work would be further suitable for the readership in the specific field of chemical biology.